# CARD: Coarse-to-fine Autoregressive Modeling with Radix-based Decomposition for Transferable Free Energy Estimation

**Ziyang Yu** [1 2 3 * †]  **Yi He** [1 *]  **Wenbing Huang** [4 5]  **Wen Yan** [1]  **Yang Liu** [2 3]

## Abstract

Estimating free energy differences quantifies thermodynamic preferences in molecular interactions, which is central to chemistry and drug discovery. Despite fruitful progress, existing methods still face key limitations: classical computational approaches remain prohibitively expensive due to their reliance on extensive molecular dynamics simulations, while deep learning-based methods are constrained by either less-expressive generative models or input dimensions tied to a specific system, resulting in negligible generalization. To address these challenges, we propose CARD, a generative framework that employs a novel radix-based decomposition to bijectively convert 3D coordinates into mixed discrete-continuous sequences, enabling coarse-to-fine autoregressive modeling with enhanced expressiveness. Notably, the model corresponds to a distribution with zero free energy, serving as a proposal for absolute free energy computation of arbitrary systems without relying on alchemical pathways. Experiments across diverse tasks demonstrate that CARD matches the accuracy of classical computational methods on unseen systems with diverse topologies, while achieving an approximately 40-fold speedup in inference.

## 1. Introduction

Estimating free energy differences is crucial for various applications in computational chemistry and biophysics, such

---
*Equal contribution †Work done as intern at ByteDance Seed [1]ByteDance Seed [2]Department of Computer Science and Technology, Tsinghua University [3]Institute for AI Industry Research (AIR), Tsinghua University [4]Gaoling School of Artificial Intelligence, Renmin University of China [5]Beijing Academy of Artificial Intelligence, Beijing, China. Correspondence to: Yi He <heyi@bytedance.com>, Wenbing Huang <hwenbing@ruc.edu.cn>, Yang Liu <liuyang2011@tsinghua.edu.cn>.

*Proceedings of the 43rd International Conference on Machine Learning*, Seoul, South Korea. PMLR 306, 2026. Copyright 2026 by the author(s).

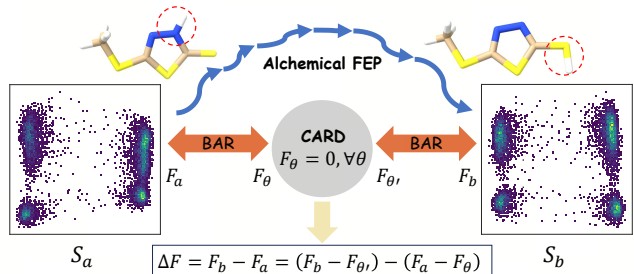

*Figure 1.* Comparison between our CARD and alchemical FEP (Bash et al., 1987) in estimating free energy differences. Alchemical FEP (blue arrows) requires multiple intermediate states and extensive MD simulations, whereas CARD acts as a zero-free-energy proposal, directly computing the absolute free energies of the two systems via BAR (Bennett, 1976).

as predicting protein-ligand binding affinities, determining solvation free energies, and mapping conformational landscapes of macromolecules (Chodera et al., 2011; Cournia et al., 2017; Chipot, 2023; Ghidini et al., 2025). Specifically, for systems $S_a$ and $S_b$ with energies $U_a$ and $U_b$, the free energy difference is expressed as

$$\Delta F = F_b - F_a = -\beta^{-1} \log \frac{Z_b}{Z_a}, \quad (1)$$

where $\beta$ is the inverse temperature, $Z_a$ and $Z_b$ correspond to the partition functions of the normalized densities of systems $S_a$ and $S_b$, respectively.

Solving Equation (1) using computational methods is a longstanding and classical problem. The well-known Free Energy Perturbation (FEP) method (Zwanzig, 1954) uses importance sampling to express the relative free energy as an exponential average of potential energy differences over a reference system. However, FEP remains limited by unstable estimates for systems with poor overlap between their distributions (Jarzynski, 2006). To address this, advanced methods like alchemical FEP (Bash et al., 1987; Mey et al., 2020) and Thermodynamic Integration (TI) (Kirkwood, 1935) introduce intermediate states to ensure sufficient phase-space overlap between neighboring systems. Alternatively, the Bennett Acceptance Ratio (BAR) (Bennett, 1976) and its multistate generalization, MBAR (Shirts & Chodera, 2008), provide minimal-variance free energy

estimators, while the Jarzynski equality (Jarzynski, 1997; Vaikuntanathan & Jarzynski, 2008) enables estimators from non-equilibrium trajectories. Despite their improvements, these classical methods require extensive sampling through Molecular Dynamics (MD) simulations, making them computationally expensive and limiting their practical applicability to large-scale systems.

Recent advancements in deep learning provide new opportunities to accelerate free energy calculations. These methods are primarily inspired by classical frameworks, including targeted FEP (Wirnsberger et al., 2020; Erdogan et al., 2025), TI (Máté et al., 2024; 2025), and Jarzynski-based approaches (Du et al., 2025). Notably, DeepBAR (Ding & Zhang, 2021) introduces a novel paradigm for free energy estimation using deep neural networks, referred to as **zero-free-energy proposals**. The key idea is to leverage tractable probabilistic models to construct a proposal distribution with zero absolute free energy, thereby enabling direct estimation of free energy differences without alchemical intermediates. However, these deep learning–based approaches suffer from two key limitations: (i) Methods such as DeepBAR, which require tractable likelihoods, typically rely on normalizing flows (Rezende & Mohamed, 2015) and thus exhibit limited expressiveness compared with modern generative frameworks like diffusion models (Ho et al., 2020) or autoregressive models (Brown et al., 2020; Achiam et al., 2023). (ii) Most approaches have input dimensionality tied to a specific target system, preventing generalization to systems with different numbers of atoms or topologies and necessitating system-specific retraining.

To address these limitations, we propose a novel framework built upon the concept of zero-free-energy proposals, as illustrated in Figure 1. First, we introduce a powerful generative model based on autoregressive modeling, which allows for the explicit calculation of probability densities while offering superior expressiveness. Second, our model is inherently universal: unlike existing approaches, our method accommodates molecular systems of arbitrary number of atoms and diverse topologies without the need for retraining. Specifically, we propose Coarse-to-Fine Autoregressive Modeling with Radix-based Decomposition (CARD), a transformer-based framework that efficiently processes mixed discrete-continuous sequences obtained from 3D coordinates through radix-based decomposition, enabling exact and scalable free energy estimation.

Overall, we summarize our key contributions as follows:

1. We introduce CARD, a novel autoregressive model that employs radix-based decomposition for coarse-to-fine conformation generation, serving as a zero-free-energy proposal for computing the absolute free energies of arbitrary molecular systems.

2. To the best of our knowledge, among deep learning methods derived from free energy theories, CARD is the first to demonstrate robust generalization to unseen systems, distinguishing it from existing approaches that require system-specific retraining.

3. Experiments demonstrate that CARD attains accuracy comparable to classical computational methods on diverse tasks, while delivering an approximately 40-fold speedup in inference.

## 2. Background

In this section, we introduce the key concepts relevant to free energy estimation to provide readers with a clear background. Section 2.1 covers the theoretical foundations of free energy and mainstream computational methods, while Section 2.2 reveals that tractable probabilistic models can serve as a free energy reference, bridging the gap between free energy theory and modern generative approaches. Due to space limitations, we defer a summary of the related works to Section B.

### 2.1. Free Energy Estimation in Molecular Systems

For simplicity, consider a system $S_a$ at constant temperature $T$ and fixed volume, characterized by an energy function $U_a : \Omega \to \mathbb{R}$, where $\Omega \subseteq \mathbb{R}^d$ denotes the configuration space. The corresponding Boltzmann distribution is

$$p_a(x) = \frac{\exp(-\beta U_a(x))}{Z_a}, \ x \in \Omega, \tag{2}$$

$$Z_a = \int_\Omega \exp(-\beta U_a(x)) \, \mathrm{d}x. \tag{3}$$

The Helmholtz free energy of system $S_a$ is defined as

$$F_a = -\beta^{-1} \log Z_a. \tag{4}$$

Similarly, the free energy $F_b$ of another system $S_b$ can be defined. The free energy difference between systems $S_a$ and $S_b$ is then given by Equation (1).

The Free Energy Perturbation (FEP) method (Zwanzig, 1954) employs importance sampling to transform Equation (1) into a statistically tractable quantity. Using samples drawn from a reference system, FEP estimates $\Delta F$ by computing the exponential average of reduced energy differences between the two systems:

$$\Delta F = -\beta^{-1} \log \mathbb{E}_a[\exp(-\beta(U_b - U_a))]. \tag{5}$$

Furthermore, alchemical FEP approaches (Bash et al., 1987; Mey et al., 2020) extend this concept by introducing a sequence of intermediate states that gradually transform system $S_a$ into $S_b$, thereby improving the overlap between successive distributions and enhancing convergence. These

intermediates are typically realized by interpolating the energy functions of the two systems using a coupling parameter.

From another perspective, the Bennett Acceptance Ratio (BAR) approach (Bennett, 1976) provides a statistically optimal estimator for the free energy difference by combining samples from both systems $S_a$ and $S_b$:

$$\Delta F = -\beta^{-1} \log \frac{\mathbb{E}_a[f(\beta(U_b - U_a - C))]}{\mathbb{E}_b[f(\beta(U_a - U_b + C))]} + C, \quad (6)$$

where $C$ represents an arbitrary energy offset, and the function $f$ is required to satisfy $\frac{f(x)}{f(-x)} = \exp(-x)$, with a common choice being the Fermi-Dirac distribution $f(x) = \frac{1}{1+\exp(x)}$. It has been demonstrated that choosing $C = \Delta F$ yields the minimum standard error for a given simulation time, and this choice can be updated self-consistently.

### 2.2. Probabilistic Models as Free Energy Reference

To avoid the additional computational cost of introducing intermediate states, DeepBAR (Ding & Zhang, 2021) leverages normalizing flows to define the free energy reference point, enabling the direct computation of absolute free energies for a given system. We show that the property holds for all tractable probabilistic models: given a learned density $q_\theta : \Omega \to \mathbb{R}_+$ parametrized by $\theta$, if we define the system's energy as $U_\theta = -\log q_\theta$, the corresponding free energy can be expressed as:

$$F_\theta = -\log \int_\Omega \exp(-U_\theta(x)) \, dx = -\log \int_\Omega q_\theta(x) \, dx = 0. \quad (7)$$

Hence, any model constructed in this manner defines a zero free-energy reference, independent of its parameters. Consequently, for a molecular system $S$ with target density $p$, once the proposal distribution $q_\theta$ is aligned with $p$ to ensure sufficient overlap, the absolute free energy of $S$ reduces to the free energy difference between $q_\theta$ and $p$. This difference can be computed directly using the BAR method, without the need for alchemical intermediates.

## 3. Method

In this section, we present the overall framework of our proposed model, as illustrated in Figure 2. We begin in Section 3.1 by introducing the necessary definitions, notations, and formal problem formulation. Building on this, Section 3.2 outlines the complete workflow of our method, CARD, emphasizing the key components and their motivations. Section 3.3 then provides a detailed description of the model architecture, with a focus on the design of the novel attention mechanism tailored to our task. Finally, Section 3.4 describes the training objectives employed.

### 3.1. Notations

In the follow-up sections, we present the proposed method with the subsequent notations. We consider a collection of molecular systems sharing identical thermodynamic conditions (*e.g.*, temperature, pressure, and solvation), denoted by $\mathcal{S} = \{S_1, S_2, \cdots\}$. Each molecular system $S = \{U, p, z, \mathcal{C}, u\} \in \mathcal{S}$ is characterized by: (i) an energy function $U : \Omega \to \mathbb{R}$ defined over the configuration space $\Omega$; (ii) the Boltzmann distribution $p \propto \exp(-\beta U)$, which in practice is represented by the empirical distribution of MD trajectories; (iii) the atomic numbers $z \in \mathbb{N}^N$ of $N$ atoms; (iv) a set of covalent bond indices $\mathcal{C}$ encoding the molecular topology; (v) a collection of $R$ reference structures $u = \{u^{(i)}\}_{i=1}^R$ randomly sampled from $p$, which encode the system's 3D geometry. For brevity, we further denote $c = \{z, \mathcal{C}, u\}$ as the context of the system.

For model architectures, $W$ and $\varphi$ (with appropriate subscripts) denote linear projections and Multi-Layer Perceptrons (MLPs), respectively. We use softmax and LN for the softmax activation and layer normalization (Ba et al., 2016). Additionally, $\mathrm{mod}$ denotes the modulo operation, $\lceil \cdot \rceil$ the ceiling function, $\langle \cdot, \cdot \rangle$ the inner product, and $\| \cdot \|_2$ the $\ell_2$-norm of a vector.

**Problem Formulation** Given a tractable probabilistic model $q_\theta(\cdot \mid c) : \Omega \to \mathbb{R}_+$, parametrized by $\theta$ and conditioned on the system context $c$, we aim to maximize the cross entropy between $p$ and $q_\theta$ over the entire collection of molecular systems:

$$\theta^* = \arg\max_\theta \mathbb{E}_{\mathcal{S}} \mathbb{E}_{p(x)} \log q_\theta(x \mid c). \quad (8)$$

The model trained in this manner is expected to generalize well to unseen systems with shared thermodynamic conditions as those in $\mathcal{S}$, given their system context $c$.

Thus, the model can be interpreted as a generalized force field for diverse molecular systems, where the energy function is given by $U_\theta : x \in \Omega \mapsto -\log q_\theta(x \mid c)$. By applying the BAR method (Equation (6)) to the parametrized force field $U_\theta$ and the force field $U$ that governs the system, we can compute the relative free energy between $q_\theta$ and $p$, which directly corresponds to the absolute free energy of the system, as shown in Section 2.2.

### 3.2. Coarse-to-fine Autoregressive Modeling

Given the rapid advancement of large language models, we adopt a transformer-based autoregressive architecture as the backbone for our tractable probabilistic model, capitalizing on its strong generalization capabilities. The detailed workflow of our proposed method is outlined below.

**Structure Alignment** For any molecular system, physical principles dictates that the energy function should satisfy

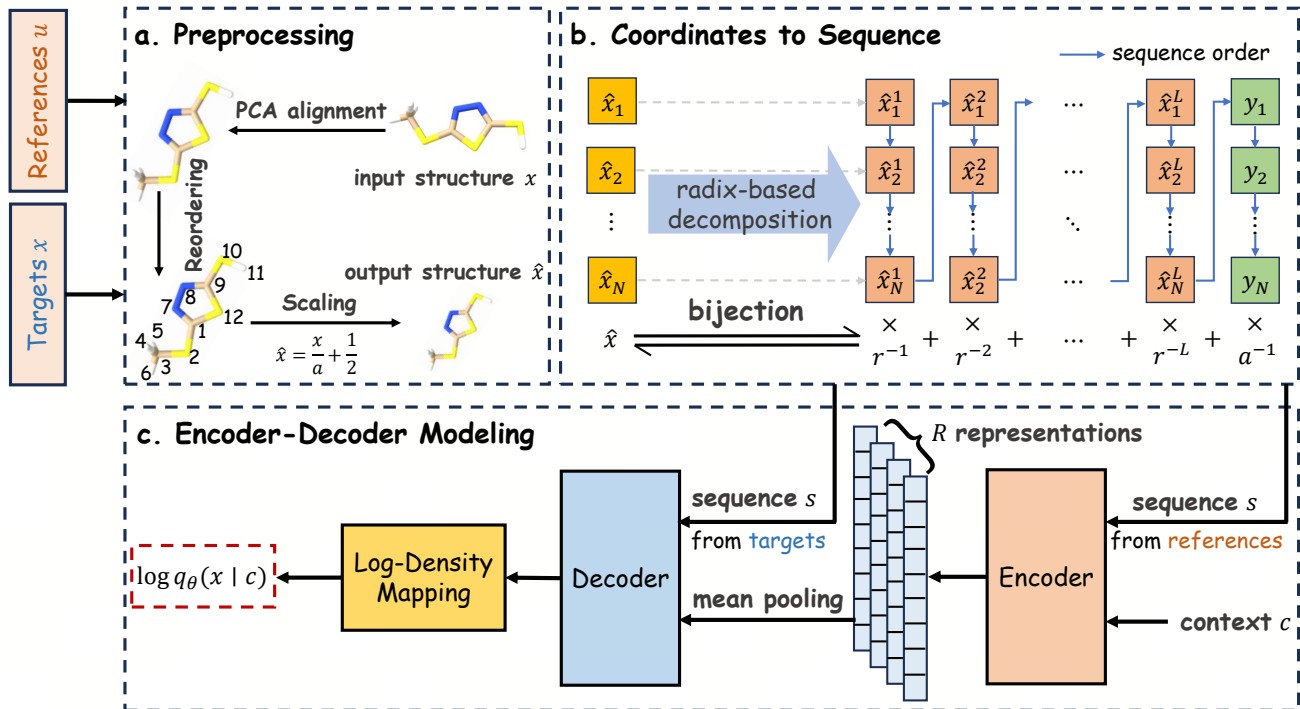

*Figure 2.* **The workflow of CARD. a.** The input structure $x$ undergoes preprocessing via PCA alignment, atom ordering, and coordinate scaling. **b.** The scaled coordinates are bijectively mapped into a mixed discrete-continuous sequence using radix-based decomposition. **c.** Reference and target sequences, along with system context $c$, are processed through a transformer-based encoder-decoder architecture to compute the log-densities $\log q_\theta(x \mid c)$.

SE(3)-equivariance, meaning it remains invariant under rotations and translations of the coordinates. To preserve this property when employing autoregressive modeling, we first align the coordinates using a Principal Component Analysis (PCA)-based procedure to eliminate rotational and translational degrees of freedom.

Specifically, each conformation is first centered by subtracting its mean across all atoms. A covariance matrix is then computed, and its three eigenvectors, ordered by decreasing eigenvalues, are used to define the principal axes. To ensure a unique orientation, the sign of the projection of the centered coordinates along each axis is checked, and axes with negative projections are flipped. The axes are further adjusted to maintain a right-handed coordinate system. Finally, the centered coordinates are rotated using these uniquely oriented axes, yielding a stable and unambiguous alignment of the 3D structure, denoted by $x \in \mathbb{R}^{N \times 3}$.

Notably, we assume the molecular conformations lack rotational symmetries, since near-degenerate principal axes in symmetric structures can lead to unstable alignments. In practice, the assumption is satisfied in the vast majority of cases.

**Autoregressive Ordering** Once the coordinates are uniquely aligned, the next step is to determine an appropriate atom ordering that simplifies the autoregressive generation while providing sufficient contextual information. In our work, we adopt two ordering strategies, depending on whether the molecular topology $\mathcal{C}$ is available.

1. **Topology-guided ordering**. This strategy is topology-driven and relies on the molecular bond graph $\mathcal{C}$. Starting from a randomly chosen atom, a depth-first search traverses the molecule, visiting neighboring atoms that covalently bonded to the current atom according to an predefined atom-type priority: carbon needs to be visited first, followed by nitrogen, then oxygen, other heavy atoms, and finally hydrogen. Random shuffling is applied among neighbors that share the same priority to introduce variation. The resulting traversal order defines the sequence in which atoms are generated.

2. **Distance-based ordering**. If $\mathcal{C}$ is not provided, the generation order is determined based on pairwise distances from the reference structures $u$. First, for each reference structure, the pairwise distances between all heavy atoms are computed, and the standard deviation of each pairwise distance across the $R$ poses is used as a cost metric. The rationale behind this is that pairs with smaller deviations are more stable in their relative positions and thus easier to generate. A heavy atom is then randomly selected as the starting point, and

the remaining heavy atoms are visited according to increasing shortest-path distances from this start atom, computed using the Floyd-Warshall algorithm (Floyd, 1962). Hydrogen atoms are placed at the end and visited in random order.

Note that the atom ordering is randomly sampled at each training iteration for data augmentation, while a fixed ordering is used during inference for each molecular system to avoid distribution shifts caused by varying orders.

**Radix-based Decomposition**  Directly generating molecular conformations in an autoregressive manner by predicting atomic coordinates sequentially poses a fundamental challenge. Each atom's position depends on all neighboring atoms, yet autoregressive modeling requires predicting an atom before its successors are known. To overcome this, we introduce a novel radix-based decomposition that bijectively converts 3D coordinates into mixed discrete-continuous sequences. This representation supports coarse-to-fine autoregressive modeling, enabling the model to capture global structures at a coarse scale before refining local details, thereby alleviating the difficulties of sequential generation. The details are presented below.

Given an aligned and reordered conformation $x \in \mathbb{R}^{N \times 3}$, we choose a sufficiently large parameter $a \in \mathbb{R}_+$ such that $|x_{ij}| < a/2$ holds for any $1 \le i \le N$, $1 \le j \le 3$. Next, we can equivalently transform $x$ into the $[0, 1)$ interval:

$$\hat{x} := \frac{x}{a} + \frac{1}{2} = \left[\hat{x}_{ij}\right]_{i=1,\dots,N}^{j=1,2,3} \in [0,1)^{N \times 3}, \qquad (9)$$

where $\hat{x}_{ij}$ denotes the $j$-th coordinate of atom $i$ after transformation, with atoms indexed according to the prescribed generation order. By specifying hyperparameters $r \in \mathbb{N}_+$ and $L \in \mathbb{N}_+$, each element of $\hat{x}$ can then be uniquely expressed in the following radix-$r$ decomposition:

$$\hat{x}_{ij} = (0.\hat{x}_{ij}^1 \hat{x}_{ij}^2 \cdots \hat{x}_{ij}^L \cdots)_r, 1 \le i \le N, 1 \le j \le 3, \qquad (10)$$

where $\hat{x}_{ij}^k \in \{0, 1, \cdots, r-1\}$ $(k \in \mathbb{N}_+)$ denotes the $k$-th digit of $\hat{x}_{ij}$ in its radix-$r$ representation. For notational convenience in the following derivations, we further define:

$$\hat{x}_i^k := \begin{bmatrix} \hat{x}_{i1}^k & \hat{x}_{i2}^k & \hat{x}_{i3}^k \end{bmatrix}^\top \in \{0, 1, \cdots, r-1\}^3, \quad (11)$$

$$y_{ij} := a \cdot (0.\underbrace{0 \cdots 0}_{L} \hat{x}_{ij}^{L+1} \cdots)_r \in [0, \frac{a}{r^L}), \qquad (12)$$

$$y_i := \begin{bmatrix} y_{i1} & y_{i2} & y_{i3} \end{bmatrix}^\top \in [0, \frac{a}{r^L})^3. \qquad (13)$$

Therefore, it is straightforward to see that each conformation $x$ is in bijective correspondence with the following mixed discrete-continuous sequence $s$:

$$s = (\hat{x}_1^1, \hat{x}_2^1, \cdots, \hat{x}_N^1, \hat{x}_1^2, \cdots, \hat{x}_N^L, y_1, \cdots, y_N). \quad (14)$$

Using the sequence $s$ defined above, atoms are refined progressively from coarse to fine scales, allowing the model to leverage the coarse spatial states of all other atoms generated in earlier steps when predicting the coordinates of the current atom. Furthermore, Proposition 3.1 shows that the log-density of any conformation $x$ under the autoregressive model $q_\theta$ can be equivalently expressed as the sum of conditional log-densities over the sequence $s$:

**Proposition 3.1.** *Given the probabilistic model $q_\theta(\cdot \mid c)$ : $\Omega \to \mathbb{R}_+$, defined over the configuration space $\Omega$ and conditioned on the system context c. For any conformation $x \in \Omega$ and its corresponding sequence s as in Equation (14), the following equality holds:*

$$\log q_\theta(x \mid c) = \sum_{i=1}^{N(L+1)} \log q_\theta(s_i \mid c, s_{:i}), \qquad (15)$$

*where $s_i$ denotes the i-th element of s, and $s_{:i} = \{s_1, \cdots, s_{i-1}\}$, with $s_{:1} = \emptyset$.*

**Beta Mixture Model**  After constructing the input sequence $s$, we next discuss how to model the conditional probability densities appearing in Equation (15).

First, For the first $NL$ elements of $s$, each admits $r^3$ possible discrete values. These conditional distributions can be parameterized using MLPs, mapping the features associated with each position to an $r^3$-dimensional output as the corresponding logits.

In contrast, the last $N$ elements $y_i$ $(1 \le i \le N)$ of $s$ are continuous values defined on a bounded interval and therefore require specialized modeling. To address this, we introduce a Beta Mixture Model (BMM) tailored for bounded continuous variables, as detailed below.

We start from the Beta distribution, whose probability density function is given by $\text{Beta}(x; \alpha, \beta) \propto x^{\alpha-1}(1-x)^{\beta-1}$ for $x \in [0, 1]$, with parameters $\alpha, \beta \in \mathbb{R}_+$. To improve expressiveness, we construct BMM by mixing $K$ Beta components with weights $\{\pi_i\}_{i=1}^K$, satisfying $\sum_{i=1}^K \pi_i = 1$. The resulting density is $\text{BMM}(x; \Theta) = \sum_{i=1}^K \pi_i \text{Beta}(x; \alpha_i, \beta_i)$, where $\Theta = \{\pi_i, \alpha_i, \beta_i\}_{i=1}^K$ represents the full set of parameters, with implementation details provided in Section E.4.

We then model each component of the target variable $y_i$ $(1 \le i \le N)$ sequentially, starting from $y_{i1}$ and conditioning subsequent components on the previously modeled ones. Each conditional distribution is represented using the above-defined BMM, which allows us to flexibly characterize the joint distribution composed of the three components of $y_i$. Finally, the expression for the conditional log-density of $y_i$ is given by Proposition 3.2:

**Proposition 3.2.** *Suppose each component of the bonded continuous variable $y_i \in [0, \frac{a}{r^L})^3$ $(1 \le i \le N)$ is modeled*

*using a BMM. By defining $c_i = \{c, s_{:i+NL}\}$ as the context of $y_i$ within the sequence $s$, the conditional log-density of $y_i$ can be expressed as:*

$$\log q_\theta(y_i \mid c_i) = \sum_{j=1}^{3} \log\left[\frac{r^L}{a} q_\theta\left(\frac{r^L}{a} y_{ij} \mid c_i, y_{i,:j}\right)\right], \quad (16)$$

*where $y_{i,:j} = \{y_{i1}, \cdots, y_{ij-1}\}$, with $y_{i,:1} = \emptyset$.*

### 3.3. Model Architecture

In this section, we mainly discuss the novel attention mechanism specifically designed for our task, which fully exploits the coarse-to-fine coordinates obtained via radix-based decomposition and the geometric information provided by the reference structures. Detailed descriptions of the full architecture, encoder-decoder workflow, and other implementation specifics are provided in Sections E and F.

**Input Expansion**   To align with the sequence $s$ defined in Equation (14), all inputs are expanded to a unified length. First, the atomic numbers $z \in \mathbb{N}^N$ are repeated $L+1$ times to form $z' \in \mathbb{N}^{N(L+1)}$. Next, we define the *decomposed coordinates* $x' \in \mathbb{R}^{N(L+1) \times 3}$ as

$$x'_{ij} = \begin{cases} a \cdot (0.\hat{x}^1_{i'j} \hat{x}^2_{i'j} \cdots \hat{x}^l_{i'j})_r - \frac{a}{2}, & i \leq NL \\ x_{i'j}, & i > NL \end{cases} \quad (17)$$

$$\text{s.t. } i' = \text{id}(i), \ l = \lceil i/N \rceil. \quad (18)$$

Here, $\text{id}(i) = ((i-1) \bmod N) + 1$ maps the sequence position $i$ to the corresponding atom index. By construction, $x'_i = \begin{bmatrix} x'_{i1} & x'_{i2} & x'_{i3} \end{bmatrix}^\top \in \mathbb{R}^3$ precisely encodes the up-to-date 3D coordinates of the atom at position $i$, with information from all subsequent positions blocked.

**Geometry-Aware Attention Mechanism**   Next, we introduce the geometry-aware multi-head attention mechanism employed in our model. Formally, let $h \in \mathbb{R}^{N(L+1) \times H}$ denote the input features for each transformer block. The query, key, and value vectors are computed as

$$q_i = (\text{LN}(h_i + \varphi_1(x'_{i-N})))W_1, \quad (19)$$

$$k_j, v_j = (\text{LN}(h_j + \varphi_2(x'_j)))W_2. \quad (20)$$

We use $x'_j$ for the keys and values to capture geometric context in real time, while incorporating $x'_{i-N}$ (set to zero if $i \leq N$) for the query prevents the coordinates $x'_i$, which are to be predicted at position $i$ during inference, from leaking into the representation.

Next, the attention weights for head $h$ are computed as

$$\alpha^h_{ij} = \text{softmax}\left(\frac{\langle q^h_i, k^h_j \rangle}{\sqrt{H_d}} + \frac{1}{R}\sum_{k=1}^{R} \varphi^h_d(d^{(k)}_{ij})\right), \quad (21)$$

$$\text{s.t. } d^{(k)}_{ij} = \|u^{(k)}_{i'} - u^{(k)}_{j'}\|_2, \ i' = \text{id}(i), \ j' = \text{id}(j). \quad (22)$$

Here, the query, key, and value vectors are split into $N_d$ attention heads, denoted by $\{q^h_i, k^h_j, v^h_j\}^{N_d}_{h=1}$, each with hidden dimension $H_d$. By incorporating $d^{(k)}_{ij}$ into the attention computation, the model explicitly leverages reference geometries to modulate attention weights, which in turn simplifies the autoregressive generation. Subsequently, the model applies standard multi-head attention using the computed attention weights, with full architectural details provided in Section E.

**Encoder-Decoder Architecture**   The encoder and decoder in our model are each stacked with $T$ identical transformer blocks. The encoder transforms atomic numbers and reference structures into geometry-aware representations. These representations, first aggregated across reference structures through mean pooling, are then fed into the decoder, which produces exact probability densities for target structures during training or generates new conformations during inference. Algorithms of training and inference procedures are further provided in Section F.

### 3.4. Training Objective

The total training objective combines a negative log-likelihood term $\mathcal{L}_{\text{NLL}}$ and an energy-alignment term $\mathcal{L}_{\text{energy}}$. First, for a batch of structures $\{x^{(b)}\}^B_{b=1}$ of size $B$, the negative log-likelihood is

$$\mathcal{L}_{\text{NLL}} = -\frac{1}{BN}\sum_{b=1}^{B} \log q_\theta(x^{(b)} \mid c). \quad (23)$$

Next, let the predicted energies be $U^{(b)}_\theta = -\log q_\theta(x^{(b)} \mid c)$ and the reference energies from the force field be $U^{(b)}$. Since only the relative differences between conformations are meaningful for energies, we perform mean-centering on both sets within each mini-batch, yielding $\tilde{U}^{(b)}_\theta$ and $\tilde{U}^{(b)}$. The energy-alignment loss is then computed as

$$\mathcal{L}_{\text{energy}} = \frac{1}{B}\sum_{b=1}^{B} \left|\tilde{U}^{(b)}_\theta - \tilde{U}^{(b)}\right|. \quad (24)$$

Finally, the total loss combines the two terms with positive weights $\lambda_1, \lambda_2 \in \mathbb{R}_+$: $\mathcal{L} = \lambda_1 \mathcal{L}_{\text{NLL}} + \lambda_2 \mathcal{L}_{\text{energy}}$.

### 3.5. Evaluation

For any two systems $S_a$ and $S_b$, the relative free energy $\Delta F$ is defined as the difference between their absolute free energies, $F_a$ and $F_b$. To compute these absolute free energies, we treat CARD as a zero-free-energy proposal according to Section 2.2, with energy defined as $U_\theta(x) = -\log q_\theta(x \mid c)$, $x \in \Omega$. The free energy difference between the proposal distribution parameterized

by $\theta$ and the target distribution is then estimated using MBAR (Shirts & Chodera, 2008), following the `pymbar` implementation[1] . For the MBAR analysis, we utilize samples from both distributions: 2,000 independent conformations are generated by the trained model following Algorithm 4, while target samples are obtained by subsampling and decorrelating the MD trajectories using `pymbar`. Given the proposal's absolute free energy is zero, the resulting MBAR estimate directly yields the absolute free energy of the target system.

## 4. Experiment

In this section, we assess CARD across diverse tasks, covering solvation free energies from vacuum to solvents (Section 4.1), endstate corrections from a classical force field to a neural network potential (Section 4.2), and aqueous tautomer free energies (Section 4.3). Additional experimental details and hyperparameter settings are provided in Section F, and ablation studies that demonstrate the effectiveness of the training strategy and the coarse-to-fine modeling scheme are shown in Section G.1.

### 4.1. Solvation Free Energy

**Setup** We first benchmark our method on the classical task of estimating solvation free energies of molecular systems from vacuum to a solvent. We select two solvents, toluene and water, to validate the model's applicability across different environments. Force fields for the different environments are set up in `OpenMM` (Eastman et al., 2024) using GAFF parameters (Wang et al., 2004), where solvent effects are approximated via the GB-OBC2 (Onufriev et al., 2000) implicit solvent model by tuning the solvent dielectric constant to 2.38 for toluene and 80.1 for water.

**Dataset** The training and test sets for this task are extracted from ZINC20 (Irwin et al., 2020), from which we uniformly sample 40,403 molecular systems across different tranches, yielding a set diverse in molecular weight and logP and restricted to electrically neutral molecules. These molecules are then randomly split into 40,203 for training and 100 each for validation and testing. To ensure chemical distinctness, we filtered the test set against the training set using a maximum Tanimoto similarity of 0.65 with Morgan ECFP4 fingerprints, yielding a final set of 70 molecules (Sayle, 2019). For all molecular systems, 10 ns MD simulations were performed in `OpenMM` under vacuum, toluene, and water force fields with a 1 ps sampling interval, collecting 10,000 conformations per thermodynamic condition. Three separate models were then trained on these samples, each fitting the empirical distribution of its respective condition.

[1]https://github.com/choderalab/pymbar

**Experimental Results** We evaluate the discrepancy between model-predicted solvation free energies on the test set and reference values obtained from Multistate Equilibrium Free Energy Simulations (MFES), following the protocol of Tkaczyk et al. (2024). The evaluation results on the 70 test systems are presented in Figure 3. In both vacuum-to-toluene and vacuum-to-water settings, the model's predictions exhibit mean absolute errors well below the 1 kcal/mol benchmark and achieve $R^2$ values exceeding 0.9, demonstrating strong agreement with MFES and robust generalization across different environments.

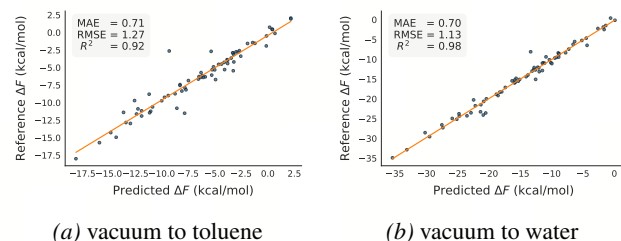

*(a)* vacuum to toluene      *(b)* vacuum to water

*Figure 3.* Comparison between model estimates ($x$-axis) and reference values ($y$-axis) for solvation free energies $\Delta F$ from *(a)* vacuum to toluene and *(b)* vacuum to water.

**Inference Efficiency** We compare the inference efficiency of CARD and MFES on 70 test systems using a single NVIDIA Tesla V100 SXM2 32GB GPU, with the results summarized in Table 1. Notably, CARD achieves an approximately 40-fold speedup over MFES when excluding the two endstate MD simulations, and maintains a 5-fold acceleration even when this simulation time is factored in. Given that endstate MD sampling remains a shared prerequisite for modern free energy estimators, this 40-fold acceleration in the actual inference phase represents a genuine leap in computational efficiency, drastically reducing the bottleneck specific to free energy estimation.

In addition, since CARD currently employs a conventional, unoptimized transformer architecture, its runtime scales approximately quadratically with the number of atoms, which accounts for the larger relative deviation observed in its mean inference time across differently sized systems.

*Table 1.* Inference time (s) comparison between CARD and MFES on 70 test systems, reported per system on a single V100 GPU. Here, $t_{end}$ denotes the total MD simulation time accumulated across both endstates.

| Method | MFES | CARD w/o $t_{end}$ | CARD w/ $t_{end}$ |
|--------|------|-------------------|------------------|
| mean | $3.23 \times 10^4$ | $7.70 \times 10^2$ | $6.65 \times 10^3$ |
| std | $6.79 \times 10^2$ | $3.53 \times 10^2$ | $4.13 \times 10^2$ |

## 4.2. Endstate Correction

**Setup**   Next, we evaluate CARD on the endstate correction task, which aims to estimate the free energy difference from a classical Molecular Mechanics (MM) force field to a Neural Network Potential (NNP). Following Tkaczyk et al. (2024), we use the same test set of 18 molecular systems filtered from the High Penalty (HiPen) set (Kearns et al., 2019). The MM calculations use the Open Force Field 2.0.0 with unconstrained bonds (Boothroyd et al., 2023), while the NNP calculations employ the ANI-2x potential (Devereux et al., 2020). Due to the high computational cost of performing MD simulations with NNPs, we subsample 7,881 and 100 molecular systems from the curated dataset in Section 4.1 for training and validation, respectively. We ensure that the training and validation sets are disjoint from the test set to avoid data leakage.

**Experimental Results**   Figure 4 presents the evaluation results of the predicted correction free energies in comparison with the reference values obtained from MFES. On the HiPen test set, which consists of molecules that are challenging for classical MM force fields, CARD achieves a mean absolute error of 0.90 kcal/mol despite being trained on a relatively small dataset, showcasing strong transferability to these difficult cases.

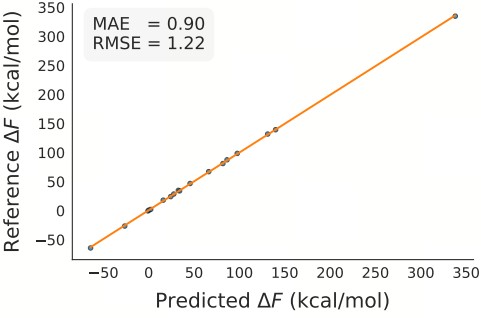

*Figure 4.* Comparison between model estimates ($x$-axis) and reference values ($y$-axis) for endstate correction free energies $\Delta F$ on 18 test systems from the HiPen set.

## 4.3. Aqueous Tautomer Free Energy

**Setup**   Finally, we present a challenging zero-shot task to estimate free energy differences between tautomers in water, where predictions are obtained using the models trained in the previous two tasks without accessing experimental free energy labels. To estimate the aqueous tautomer free energy $\Delta F_{0 \to 1}^{w}$, we decompose it into three components:

$$\Delta F_{0 \to 1}^{w} = \Delta F_{0 \to 1}^{v} + \Delta F_{1}^{v \to w} - \Delta F_{0}^{v \to w}. \quad (25)$$

Here, subscripts 0 and 1 denote the two tautomers, and superscripts $v$ and $w$ indicate vacuum and water environments,

respectively. The first term corresponds to the alchemical free energy of transforming tautomer 0 to 1 in vacuum, which we compute using the ANI-2x potential and the associated model described in Section 4.2. The remaining two terms are hydration free energies of two tautomers, which are evaluated using the models trained in Section 4.1 together with the corresponding MM force fields.

For evaluation, we use the 100-tautomer set (Pan et al., 2025), excluding systems that cannot be modeled with ANI-2x or have experimental aqueous free-energy differences exceeding 2.72 kcal/mol (corresponding to one tautomer dominating 99%), as such pairs provide limited insight. This filtering leaves 27 test molecules with experimental tautomer free energies as references.

*Table 2.* Statistics of CARD and baseline methods evaluated on 27 test tautomer pairs. MAE and RMSE are reported in kcal/mol. PCC and SPCC denote the Pearson and Spearman rank correlation coefficients, respectively. The best performance for each metric is highlighted in **bold**.

| Method | MAE ($\downarrow$) | RMSE ($\downarrow$) | PCC ($\uparrow$) | SPCC ($\uparrow$) |
|---|---|---|---|---|
| DFT | 4.62 | 7.05 | 0.36 | 0.42 |
| sPhysNet-pre | 4.61 | 6.95 | 0.35 | 0.41 |
| CARD (*ours*) | **4.11** | **5.49** | **0.64** | **0.64** |

**Experimental Results**   We compare CARD with Density Functional Theory (DFT) at the B3LYP/6-31G* level using the universal solvation model (SMD), as well as with sPhysNet (Pan et al., 2025), a regression-based method pretrained on approximately 100 million MMFF94-optimized geometries with DFT-calculated energies (denoted sPhysNet-pre). Baseline evaluation results are taken from Pan et al. (2025).

Results in Table 2 show that, despite inaccuracies inherent in classical force fields relative to DFT, CARD consistently outperforms the baselines across all statistical metrics. This can be attributed to the strong theoretical foundation of CARD, which ensures that free energy estimates are free from intrinsic methodological bias, with remaining errors arising only from model or force-field imperfections. In contrast, DFT approximates free energy differences using the energy difference between minimum-energy conformations of the two tautomers, a simplification of Equation (1) that often leads to large deviations in complex systems.

To further substantiate this observation, we visualize the results in Figure 5, grouping tautomer pairs by their mean rotatable-bond count as a proxy for molecular flexibility and complexity. We observe that while DFT outperforms CARD on simpler systems, it shows larger errors and outliers as system complexity increases. In contrast, CARD manifests enhanced stability, highlighting its consistent robustness across varying system complexities.

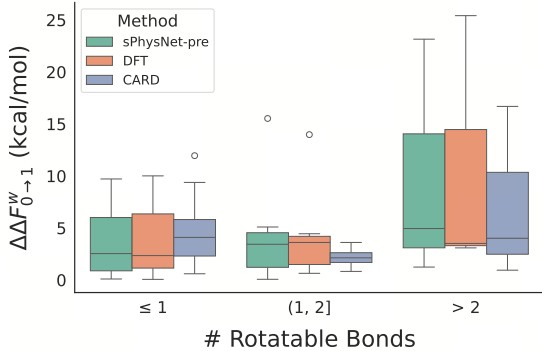

*Figure 5.* Comparison of CARD with baselines on the 27 test pairs, grouped by the mean rotatable-bond count of the two tautomers. $\Delta\Delta F_{0\to1}^{w}$ on the $y$-axis denotes the absolute error between predicted and experimental values in kcal/mol.

### 4.4. Ablation Study

To assess the impact of key hyperparameters (*i.e.*, radix $r$ and depth $L$) on model performance, as well as the necessity of the dual training objectives introduced in Section 3.4, we conducted ablation studies on the vacuum-to-toluene solvation free energy task. The evaluation results are demonstrated in Table 4. Specifically, the *Training stage* column in the table refers to our proposed two-stage training strategy: the first stage solely optimizes the log-likelihood $\mathcal{L}_{\mathrm{NLL}}$, while the second stage jointly optimizes both training objectives. Further details are provided in Section F.1.

**Radix $r$** As shown in Table 3, a radix of $r = 4$ yields the optimal performance when other parameters are held constant. This reflects a critical trade-off: for a fixed depth $L$, a smaller radix forces the BMM to model a broader range of residuals, potentially exceeding its representational capacity. Conversely, a larger radix expands the discrete representation space cubically, substantially increasing the modeling complexity.

**Depth $L$** Fixing the radix at $r = 4$, we subsequently evaluated the impact of depth $L$. As shown in Table 3, reducing the depth to $L = 2$ causes a significant performance drop compared with $L = 3$, underscoring the necessity of the coarse-to-fine modeling scheme to provide sufficient contextual information. Conversely, increasing the depth to $L = 4$ leads to a slight performance degradation. This is likely because, at higher depth level $k$, the decomposed discrete variables $\hat{x}_i^k$ ($1 \leq i \leq N$) become increasingly indistinguishable among the $r^3$ classes, making it more difficult for the model to extract informative signals.

**Two-stage Training** We further investigate the impact of the introduced two-stage training on the performance. Under the optimal parameter setting ($r = 4$ and $L = 3$), incorporat-

ing Stage II during training markedly improves performance across all metrics. This improvement is likely attributable to the use of ground-truth force field energy labels, which effectively corrects the biases in the parametrized distribution $q_\theta$ arising from inadequate conformational sampling, such as imbalanced data distribution and incomplete phase space coverage in finite MD trajectories.

*Table 3.* Ablation study on the effect of the radix $r$, depth $L$, and two-stage training on model performance, evaluated on the solvation free energy task from vacuum to toluene. Pct ($<1$) denotes the fraction of predictions whose absolute error is smaller than 1 kcal/mol. MAE and RMSE are reported in kcal/mol. The best result for each metric is shown **bold**.

| Training stage | Parameters | MAE | RMSE | $R^2$ | Pct ($<1$) |
|---|---|---|---|---|---|
| Stage I & II | $r = 4, L = 3$ | **0.71** | **1.27** | **0.92** | **82.9** |
| Stage I | $r = 4, L = 3$ | 0.81 | 1.34 | 0.91 | 77.1 |
| | $r = 3, L = 3$ | 2.43 | 3.08 | 0.61 | 26.5 |
| | $r = 5, L = 3$ | 1.88 | 2.41 | 0.73 | 22.1 |
| | $r = 4, L = 2$ | 5.85 | 14.26 | -0.08 | 17.1 |
| | $r = 4, L = 4$ | 1.43 | 2.39 | 0.77 | 61.4 |

## 5. Conclusion

Free energy estimation is fundamental to understanding thermodynamic preferences in molecular interactions. In this work, we propose a powerful generative framework, namely CARD, which processes a mixed discrete-continuous sequence derived from 3D coordinates via a novel radix-based decomposition, enabling coarse-to-fine autoregressive modeling with enhanced expressiveness. Grounded in solid theoretical principles, CARD provides a zero-free-energy proposal for directly computing absolute free energies of arbitrary systems. Notably, experiments across multiple tasks demonstrate its robust generalization to unseen systems with varying topologies, marking a significant breakthrough in deep learning-based free energy estimation.

There remains room for improvement. First, CARD currently removes roto-translational degrees of freedom using PCA, which may introduce large variance for symmetric conformations. More robust invariant feature constructions are needed. Second, our evaluation focuses on drug-like molecules, and the extension to larger systems such as protein-ligand complexes remains an open direction.

## Acknowledgements

This work was jointly supported by the Fundamental and Interdisciplinary Disciplines Breakthrough Plan of the Ministry of Education of China (No. JYB2025XDXM101), the Beijing Major Science and Technology Project (No. Z251100008125061), and Beijing Academy of Artificial Intelligence (BAAI).

## Impact Statement

Accurate and efficient estimation of free energy differences remains a central challenge in molecular science, with far-reaching consequences for drug discovery, materials development, and chemical engineering. In this work, we propose CARD, a deep learning framework rooted in free energy theories that enables absolute free energy estimation without relying on alchemical pathways or system-specific retraining. By introducing a novel radix-based decomposition for coarse-to-fine autoregressive modeling, CARD exhibits strong generalization to unseen molecular systems under shared thermodynamic conditions.

From a practical standpoint, CARD significantly reduces the computational burden of free energy calculations. In our experiments, the framework achieves approximately a $40\times$ speedup in inference compared to classical computational methods with comparable accuracy. This improvement substantially lowers the cost of large-scale virtual screening and facilitates rapid exploration of chemical space, which is particularly valuable in early-stage drug discovery.

Furthermore, this work also highlights the benefits of integrating physical principles with modern deep learning architectures, which addresses long-standing limitations of both purely data-driven and classical computational methods. Specifically, CARD provides a tractable distribution that generalizes well to unseen systems given their system contexts, thereby supporting reliable downstream free energy estimation through efficient parallel sampling.

Overall, we expect CARD to contribute positively to the field by enabling faster, more scalable, and physically grounded approaches to molecular free energy estimation.

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

## Appendix

## A. Reproducibility

To facilitate reproducibility, the source code is publicly available at `https://github.com/bytedance/CARD`.

## B. Related Work

**Computational Methods for Free Energy Estimation**    Accurate estimation of free energies poses a central challenge in physics and biochemistry, which cannot be evaluated directly due to the need to adequately sample rare but important regions of phase space. Therefore, classical computational approaches have been developed to provide numerical estimates with controlled accuracy. First, the well-known Free Energy Perturbation (FEP) method expresses the relative free energy between two states as an exponential average of potential energy differences sampled from a reference ensemble according to the Zwanzig equation (Zwanzig, 1954). Due to reliance on importance sampling, FEP requires significant overlap between the probability distributions of the two states, while insufficient overlap leads to highly unstable and high-variance estimates (Jarzynski, 2006).

To improve convergence further, advanced formulations of FEP like alchemical FEP (Bash et al., 1987; Mey et al., 2020) interpolate the Hamiltonian between the states via a series of discrete intermediates, ensuring ensembles of adjacent states overlap sufficiently. Thermodynamic Integration (TI) (Kirkwood, 1935) takes a related but continuous approach by defining a coupling parameter that interpolates between the reference and target Hamiltonians, where the free energy difference is computed by integrating the ensemble-averaged derivative of the Hamiltonian with respect to this parameter along the path. Subsequently, targeted FEP (Jarzynski, 2002) introduces a high-dimensional invertible mapping to enlarge the effective overlap between the mapped reference distribution and the target distribution, without additional sampling from intermediate states. From another angle, Bennett Acceptance Ratio (BAR) (Bennett, 1976) further refines estimation by combining samples from both end states to provide a minimum-variance estimator, and its multistate generalization, MBAR (Shirts & Chodera, 2008), extends this idea to multiple intermediate states, yielding statistically optimal estimates across all windows simultaneously. Meanwhile, non-equilibrium methods based on the Jarzynski equality allow free energy differences to be estimated from ensembles of non-equilibrium trajectories (Jarzynski, 1997; Vaikuntanathan & Jarzynski, 2008). Despite their improvements in convergence and accuracy, these methods require extensive sampling through molecular dynamics or Monte Carlo simulations to achieve equilibrated ensembles or sufficient overlap across states, which are computationally expensive and limits their practical applicability in large or complex systems.

**Deep Learning for Free Energy Estimation**    Rapid progress in deep learning, especially in generative models, opens up new avenues for free energy estimation. Deep learning-based methods fall into two main categories: (i) data-driven approaches trained with free energy labels, and (ii) theory-based methods grounded in classical computational frameworks. Methods in the first category primarily take 3D molecular conformations as input and directly yield scalars as free energy predictions, addressing tasks such as protein-protein interactions prediction (Romero-Molina et al., 2022; Zou et al., 2024; Xie et al., 2025; Zhang et al., 2025), protein-ligand binding affinity prediction (Karimi et al., 2019; Townshend et al., 2021; Liu et al., 2023; Zhou et al., 2023; Gao et al., 2024; Feng et al., 2024; Li & Gong, 2025), aqueous tautomer ratio estimation (Wieder et al., 2021; Pan et al., 2023; 2025), etc. These data-driven methods, while highly efficient, often struggle to generalize to unseen and complex molecular systems with non-trivial regions of configuration space missing from training data (Duschatko et al., 2024; Mendible et al., 2024).

From another perspective, methods in the second category incorporates generative models into classical frameworks, striking a balance between the expressiveness of deep learning approaches and the reliability of established free energy theories. Building on targeted FEP (Jarzynski, 2002), some methods learn invertible transports that push reference configurations toward the high-probability regions of the target distribution (Wirnsberger et al., 2020; Erdogan et al., 2025). In parallel, neural thermodynamic integration (Máté et al., 2024; 2025) leverages energy-based diffusion models (Song et al., 2021) to parameterize a time-dependent Hamiltonian that interpolates between the reference and target systems, where TI (Kirkwood, 1935) can be performed using the score from the learned process. FEAT (Du et al., 2025) further unifies equilibrium and non-equilibrium estimators through adaptive stochastic transports (Albergo et al., 2025). Notably, DeepBAR (Ding & Zhang, 2021) employs normalizing flows (Rezende & Mohamed, 2015) to construct a reference distribution with an inherent zero free energy, which serves as a proposal for directly estimation of absolute free energies and leads to a fundamentally new paradigm. However, these theory-based approaches suffer from a critical limitation: they require system-specific

retraining with input dimensions tied to a specific system, thereby failing to generalize to unseen systems and substantially undermining their practical advantage over classical approaches.

**Molecular Ensemble Generation**    Free energy estimation relies on adequate sampling of the configuration space, a requirement that can be naturally addressed by generative models through efficient ensemble generation. One alternative is to run MD simulations with neural network potentials, which achieve *ab initio* accuracy for substantially larger systems, but remain limited by inherently sequential sampling (Wang et al., 2024a). Mainstream approaches leverage modern generative models, such as diffusion (Ho et al., 2020; Song et al., 2021) and flow matching (Lipman et al., 2023), to learn the empirical distribution from MD trajectories and enable parallel sampling (Jing et al., 2024; Wang et al., 2024b; Lewis et al., 2025). Moreover, *Boltzmann generators* (Noé et al., 2019; Klein & Noé, 2024; Tan et al., 2025b) treat trained generative models as surrogates that are used to resample and reproduce the Boltzmann ensemble. Although parallel sampling substantially improves efficiency, most of these methods either cannot efficiently compute the log-likelihoods of generated samples or fail to generalize across different systems, and therefore are not applicable to the zero-free-energy proposal paradigm for free energy estimation.

**Multiresolution Molecular Modeling**    Recent advancements in graph representation learning have increasingly adopted multiresolution strategies to capture the inherent hierarchical nature of molecular structures. For instance, MGVAE (Hy & Kondor, 2023) introduces a hierarchical generative framework that applies higher-order message passing to progressively partition and coarsen graphs, yielding an equivariant hierarchy of latent distributions. Similarly, MGT (Ngo et al., 2023) learns macromolecular representations across multiple scales by iteratively grouping atoms into functional substructures, effectively modeling both local interactions and global topology. The Sequoia framework (Trang et al., 2024) also proposes an adaptive hierarchical self-attention mechanism that dynamically constructs a data-dependent hierarchy, which significantly reduces computational complexity while preserving the capacity to model long-range dependencies. Furthermore, GET (Kong et al., 2024) introduces a unified framework capable of encoding diverse molecular structures by conceptualizing any 3D complex as a geometric graph of building blocks, enabling the representation of various molecular modalities within a single model.

Collectively, these methods rely on hierarchical graph architectures to learn molecular substructures, primarily to improve attention efficiency and scalability. In contrast, rather than operating on abstracted graph topologies, CARD performs multi-level modeling directly on fine-grained, all-atom 3D coordinates. This coordinate-level decomposition addresses the critical challenge of prematurely committing to local coordinates before global geometry is established. Importantly, it strictly preserves exact autoregressive factorization and tractable likelihood evaluation, distinguishing CARD from prior hierarchical graph-based approaches.

# C. Proof of Propositions

## C.1. Proof of Proposition 3.1

Assume that every conformation $x \in \Omega$ admits a unique PCA-based structural alignment as described in Section 3.2, which removes the roto-translational degrees of freedom and yields invariant features for prediction. By choosing a sufficient large constant $a \in \mathbb{R}_+$ such that the aligned coordinates $x \in [-\frac{a}{2}, \frac{a}{2})^{N \times 3}$, we define a mapping $f : \{0, 1, \cdots, r-1\}^L \times [0, \frac{a}{r^L}) \to [-\frac{a}{2}, \frac{a}{2})$:

$$f(u) = a \sum_{l=1}^{L} u_l \cdot r^{-l} + u_{L+1} - \frac{a}{2} \in [-\frac{a}{2}, \frac{a}{2}), \tag{26}$$

$$\text{s.t. } u := \begin{pmatrix} u_1 & u_2 & \cdots & u_{L+1} \end{pmatrix}, \ u_1, u_2, \cdots, u_L \in \{0, 1, \cdots, r-1\}, \ u_{L+1} \in [0, \frac{a}{r^L}). \tag{27}$$

We first show that $f(u)$ is bounded within $[-\frac{a}{2}, \frac{a}{2})$:

$$f(u) \geq a \sum_{l=1}^{L} 0 \cdot r^{-l} + 0 - \frac{a}{2} = -\frac{a}{2}, \ f(u) < a \sum_{l=1}^{L} (r-1) r^{-l} + \frac{a}{r^L} - \frac{a}{2} = a(\sum_{l=1}^{L} (r-1) r^{-l} + r^{-L}) - \frac{a}{2} = \frac{a}{2}. \tag{28}$$

Next, we prove that $f$ is injective. For any $u \neq v$, let $k$ be the smallest index such that $u_k \neq v_k$. Without loss of generality, assume that $u_k < v_k$. We distinguish two cases: (i) if $k = L + 1$, then it is immediate that $f(u) < f(v)$; (ii) if $k \leq L$, then

since $u_k + 1 \le v_k$, we have

$$f(u) = a \sum_{l=1}^{L} u_l \cdot r^{-l} + u_{L+1} - \frac{a}{2} = a \left( \sum_{l=1}^{k-1} v_l \cdot r^{-l} + u_k \cdot r^{-k} + \sum_{l=k+1}^{L} u_l \cdot r^{-l} \right) + u_{L+1} - \frac{a}{2} \tag{29}$$

$$< a \left( \sum_{l=1}^{k-1} v_l \cdot r^{-l} + u_k \cdot r^{-k} + \sum_{l=k+1}^{L} (r-1) r^{-l} \right) + \frac{a}{r^L} - \frac{a}{2} \tag{30}$$

$$= a \left( \sum_{l=1}^{k-1} v_l \cdot r^{-l} + (u_k + 1) \cdot r^{-k} \right) - \frac{a}{2} \tag{31}$$

$$\le a \left( \sum_{l=1}^{k-1} v_l \cdot r^{-l} + v_k \cdot r^{-k} \right) - \frac{a}{2} \le f(v). \tag{32}$$

In both cases, we have $f(u) < f(v)$, thus $f$ is an injective mapping.

Further, we prove that $f$ is surjective. For any continuous variable $c \in [-\frac{a}{2}, \frac{a}{2})$, since $\hat{c} := \frac{c}{a} + \frac{1}{2} \in [0, 1)$, we can convert it into a purely fractional representation in base-$r$: $\hat{c} = (0.\hat{c}_1 \hat{c}_2 \cdots \hat{c}_L \cdots)_r$, $\hat{c}_i \in \{0, 1, \cdots, r-1\}$, $i \in \mathbb{N}_+$. By defining

$$u_l = \begin{cases} \hat{c}_l, & 1 \le l \le L, \\ a \cdot (0.\underbrace{0 \cdots 0}_{L} \hat{c}_{L+1} \cdots)_r, & l = L+1. \end{cases} \tag{33}$$

It is straightforward to verify that $u = \begin{pmatrix} u_1 & u_2 & \cdots & u_{L+1} \end{pmatrix}$ satisfies $f(u) = c$. Therefore, $f$ is surjective, and since we have already shown that $f$ is injective, it follows that $f$ is a bijection.

Finally, we apply the bijection $f^{-1}$ to the aligned coordinates $x$. For each atom $i = 1, \ldots, N$ and each Cartesian component $j = 1, 2, 3$, define the mixed discrete-continuous sequence

$$u^{(i,j)} = f^{-1}(x_{ij}) \in \{0, 1, \ldots, r-1\}^L \times [0, \frac{a}{r^L}). \tag{34}$$

Concatenate the 3-dimensional sequences for each atom:

$$U^{(i)} = \begin{pmatrix} u^{(i,1)}, u^{(i,2)}, u^{(i,3)} \end{pmatrix}, \tag{35}$$

and then concatenate all $N$ sequences:

$$U = \begin{pmatrix} U^{(1)}, \ldots, U^{(N)} \end{pmatrix}. \tag{36}$$

Finally, apply a permutation $\pi$ to reorder the sequences so that they match the form given in Equation (14). We then define the overall mapping

$$g^{-1} : x \mapsto s = \pi(U) \in (\{0, 1, \ldots, r-1\}^L \times [0, a/r^L))^{3N}. \tag{37}$$

By construction, $g^{-1}$ is a bijection between the bounded Euclidean space $[-\frac{a}{2}, \frac{a}{2})^{N \times 3}$ and the mixed discrete-continuous space $(\{0, 1, \ldots, r-1\}^L \times [0, a/r^L))^{3N}$.

Let $s = (s_d, s_c)$ denote the mixed sequence obtained from $g^{-1}(x)$, where $s_d$ are the discrete digits and $s_c$ the continuous components. Then, by the change-of-variables formula, we have

$$q_X(x \mid c) = \sum_{s \in g^{-1}(x)} q_S(s \mid c) \left| \det \frac{\partial s_c}{\partial x} \right|, \tag{38}$$

where $g^{-1}(x)$ denotes all possible sequences mapping to $x$. Since $g^{-1}$ is a bijection, there is a unique sequence $s = g^{-1}(x)$ corresponding to each $x$. Moreover, the Jacobian of the continuous components, $\partial s_c / \partial x$, is an identity matrix because $\partial u_{L+1}^{(i,j)} / \partial f = 1$. Therefore, its determinant satisfies $\left| \det \frac{\partial s_c}{\partial x} \right| = 1$. Thus, in our construction, the density reduces to

$$q_X(x \mid c) = q_S(s \mid c) = \prod_{i=1}^{N(L+1)} q_\theta(s_i \mid c, s_{:i}), \tag{39}$$

with $s = g^{-1}(x)$. This concludes the proof. $\square$

## C.2. Proof of Proposition 3.2

For each atom $i = 1, \ldots, N$ and each Cartesian component $j = 1, 2, 3$, we have $y_{ij} \in [0, a/r^L)$. To place it within the domain of the BMM, we introduce the scaled variable

$$y'_{ij} := \frac{r^L}{a} \, y_{ij} \in [0, 1), \tag{40}$$

which lies in $[0, 1)$ and can therefore be directly modeled by the proposed BMM.

Applying the change-of-variables formula, we obtain

$$q_Y(y_{ij} \mid c_i, y_{i,:j}) = q_{Y'}(y'_{ij} \mid c_i, y_{i,:j}) \left| \det \frac{\partial y'_{ij}}{\partial y_{ij}} \right| = \frac{r^L}{a} \, q_{Y'}(y'_{ij} \mid c_i, y_{i,:j}). \tag{41}$$

Finally, by the chain rule, the log-density of the full 3D coordinate $y_i$ is

$$\log q_\theta(y_i \mid c_i) = \sum_{j=1}^{3} \log q_\theta(y_{ij} \mid c_i, y_{i,:j}) = \sum_{j=1}^{3} \log \left[ \frac{r^L}{a} \, q_\theta \left( \frac{r^L}{a} y_{ij} \mid c_i, y_{i,:j} \right) \right]. \tag{42}$$

This completes the proof. $\qquad\qquad\qquad\qquad\qquad\qquad\qquad\qquad\qquad\qquad\qquad\qquad\qquad\qquad\qquad\qquad\qquad\quad \square$

# D. Dataset Construction

### D.1. Molecular Dynamics Simulation Setup

Molecular dynamics (MD) simulations were conducted to generate trajectories for each molecular system for both training and evaluation purposes. For the solvation free energy task (Section 4.1), initial conformations were generated from SMILES using RDKit (rdk) and optimized with MMFF (Halgren, 1996). The optimized structures were then converted into AMBER topologies and coordinates using GAFF parameters (Wang et al., 2004). System construction was subsequently performed using a unified protocol, with the only difference arising from the treatment of solvation. In the gas-phase (vacuum) setting, systems were constructed without an implicit solvent model, and nonbonded interactions were evaluated without a cutoff. In the implicit-solvent setting, the OBC2 generalized Born model (Onufriev et al., 2000) with the appropriate solvent dielectric constant was applied (2.38 for toluene and 80.1 for water), while all other simulation parameters were kept identical.

For the endstate correction task (Section 4.2), topologies were generated from SMILES using OpenFF (Mobley et al., 2018), and initial conformations were obtained via energy relaxation in OpenMM (Eastman et al., 2024) under the corresponding force field. In particular, Open Force Field 2.0.0 with unconstrained bonds (Boothroyd et al., 2023) and the ANI-2x potential (Devereux et al., 2020) were used for the MM and NNP cases, respectively.

In particular, the aqueous tautomer free energy task (Section 4.3) involves both vacuum and implicit-solvent simulations using MM force fields and the ANI-2x potential. Therefore, the simulation systems in each environment were constructed following the same protocols as in the corresponding tasks described above.

With the initial conformations prepared, MD simulations were performed in OpenMM at 300 K using a Langevin integrator with a friction coefficient of $1 \text{ ps}^{-1}$ and a time step of 1 fs. To balance sampling quality and computational efficiency, systems parameterized with MM force fields were simulated for a total of 10 ns, whereas simulations employing ANI-2x were restricted to 5 ns due to the substantially higher computational cost of neural network potentials. Conformations were recorded every 1 ps.

### D.2. Conformation Flexibility Analysis

To determine whether the 10 ns MD simulations are sufficient for adequate conformational sampling, we performed a focused analysis of molecular flexibility. Specifically, we randomly selected four molecules from the training set and visualized their conformational landscapes based on heavy-atom dihedral angles. For each molecule, the dihedrals were ranked by their circular variance. We then plotted Ramachandran plots for the two most and two least mobile torsions. As illustrated in Figure 6, even the least mobile torsions exhibit distinct basins, indicating that the 10 ns trajectories successfully capture substantial conformational flexibility and adequately explore the relevant phase space.

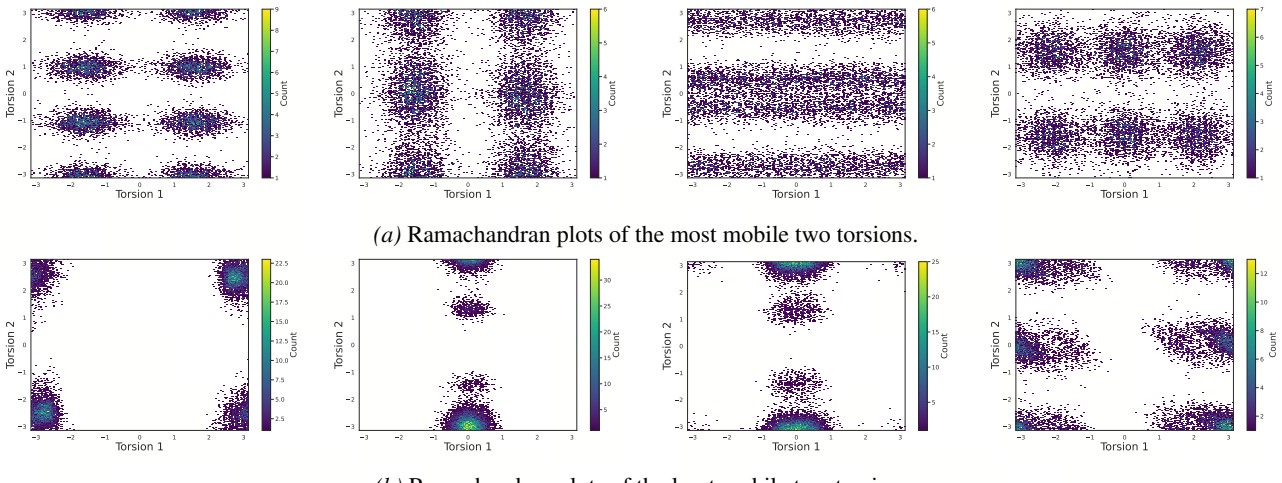

*(a)* Ramachandran plots of the most mobile two torsions.

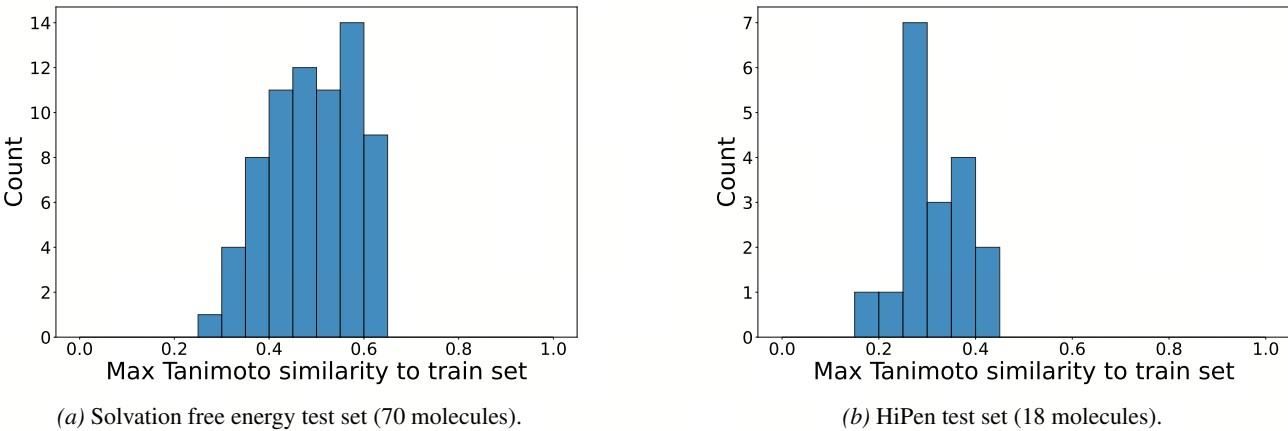

*(b)* Ramachandran plots of the least mobile two torsions.

*Figure 6.* Ramachandran plots computed from 10 ns MD trajectories of randomly selected training molecules, including ZINC000917052030 (Col. 1), ZINC000008643431 (Col. 2), ZINC000072374994 (Col. 3), and ZINC000125468547 (Col. 4).

## D.3. Tanimoto Similarity Analysis

To assess the chemical distinctness of the test sets used for solvation free energy estimation and endstate correction, we calculated the maximum Tanimoto similarity between each test molecule and the corresponding training set. Specifically, similarities were evaluated using RDKit (rdk) by parsing valid SMILES strings into 2048-bit binary Morgan (ECFP4) fingerprints (radius 2 with chirality enabled), followed by a bulk nearest-neighbor search to extract pairwise Tanimoto similarities across the bit vectors. The resulting similarity distributions are visualized as histograms in Figure 7. Both test sets maintain a maximum Tanimoto similarity below the 0.65 threshold (Sayle, 2019), confirming their chemical distinctness from the training data. Furthermore, all test cases in the HiPen set exhibit a maximum Tanimoto similarity of less than 0.5, highlighting the substantial out-of-distribution challenge posed by this specific dataset.

*(a)* Solvation free energy test set (70 molecules).

*(b)* HiPen test set (18 molecules).

*Figure 7.* Tanimoto similarity analysis of the two test sets, where each molecule is assigned its maximum similarity to the corresponding training set, estimated with ECFP4 fingerprints.

## E. Model Architecture

### E.1. Masking Strategy

The masking strategy in our model differs between the encoder and the decoder:

1. **Encoder:** operates in a bidirectional manner, with all atoms in the input sequence visible to each position. This allows the model to fully leverage the reference structures to produce geometry-aware atomic representations.

2. **Decoder:** employs a modified causal masking strategy, where for each position $i$ to be predicted, only the preceding $N$ positions, from $i - N$ to $i - 1$, are visible. We adopt this limited context because $x'_k$ ($i - N \leq k \leq i - 1$) precisely contains the up-to-date coordinates of the $N$ atoms in the system preceding position $i$, making positions earlier than $i - N$ redundant and safely maskable.

### E.2. Embeddings

Given the expanded atomic numbers $z' \in \mathbb{N}^{N(L+1) \times 3}$, the input features $h \in \mathbb{R}^{N(L+1) \times H}$ for the encoder is obtained through a mixture of embeddings:

$$h_i = \text{a\_embed}(z'_i) + \text{d\_embed}(\lceil \frac{i}{N} \rceil) \in \mathbb{R}^H, \tag{43}$$

where $H$ is the hidden dimension, a\_embed denotes the embedding of atomic numbers, and d\_embed denotes the embedding of the number of retained digits of the current coordinates in base $r$.

### E.3. Architecture of the Geometry-Aware Transformer

In summary, we present the architecture of the geometry-aware transformer block in Algorithm 1, where both the encoder and decoder are composed of $T$ identical transformer blocks stacked sequentially.

---

**Algorithm 1** Architecture of the Geometry-Aware Transformer Block

---

1: **function** MultiHeadAttention($h, x', u, \text{mask}$)
2:     $i, j \leftarrow \text{mask}$
3:     $\left[ d_{ij} \right]_{i,j=1}^{N(L+1)} \leftarrow \text{cdist}(\text{repeat}(x', L+1))$              {Getting Pairwise Distances of Reference Structures}
4:     $q_i \leftarrow (\text{LN}(h_i + \varphi_1(x'_{i-N})))W_1$
5:     $k_j, v_j \leftarrow (\text{LN}(h_j + \varphi_2(x'_j)))W_2$
6:     $\{q_i^h, k_j^h, v_j^h\}_{h=1}^{N_d} \leftarrow \text{split}(q_i, k_j, v_j; N_d)$              {Splitting into $N_d$ Heads}
7:     $\alpha_{ij}^h \leftarrow \text{softmax}_j \left( \frac{\langle q_i^h, k_j^h \rangle}{\sqrt{H_d}} + \frac{1}{R} \sum_{k=1}^{R} \varphi_d^h(d_{ij}^{(k)}) \right)$
8:     $o_i \leftarrow \text{concat}(\sum_j \alpha_{ij}^h v_j^h)_{h=1}^{N_d} W_3$
9:     **return** $o$
10: **end function**
11: **function** TransformerBlock($h, x', u, \text{mask}$)
12:     $o_1 \leftarrow \text{MultiHeadAttention}(h, x', u, \text{mask})$              {Multi-Head Attention}
13:     $h \leftarrow h + o_1$
14:     $o_2 \leftarrow \varphi_3(\text{LN}(h))$              {Pre-LayerNorm}
15:     $h \leftarrow h + o_2$
16:     **return** $h$
17: **end function**

---

### E.4. Log-density Computation

We use the decoder outputs $h_d \in \mathbb{R}^{N(L+1) \times H}$ to compute log-densities. For the first $NL$ discrete variables in $s$, each $h_d[i]$ ($1 \leq i \leq NL$) is passed through a two-layer FFN to genera the logits for $r^3$ classes at position $i$, which are then converted to log-probabilities using the log-softmax function.

Next, the remaining $N$ continuous variables $y_i$ ($1 \leq i \leq N$) are modeled using a BMM, as described in Proposition 3.2. The architecture of the BMM and the corresponding log-density computations are detailed in Algorithm 2.

---

**Algorithm 2** Architecture of the Beta Mixture Model

---

1: **function** BMM1($h$)
2:     $o \leftarrow \varphi_p(h)$          {Mapping Features to $3K$ Parameters}
3:     $\{p_i, \alpha_i, \beta_i\}_{i=1}^{K} \leftarrow \text{split}(o; K)$
4:     $\alpha_i \leftarrow \text{clamp}(\alpha_i; [-3, 3]), \; \beta_i \leftarrow \text{clamp}(\beta_i; [-3, 3])$      {Clamping the Shape Parameters for Numerical Stability}
5:     $\alpha_i \leftarrow \exp(\alpha_i), \; \beta_i \leftarrow \exp(\beta_i)$      {Exponentiation to Ensure Positivity}
6:     $p_i \leftarrow \exp(p_i)/\sum_{j=1}^{K} \exp(p_j)$      {Softmax for Mixture Probabilities}
7:     **return** BMM($\{p_i, \alpha_i, \beta_i\}_{i=1}^{K}$)
8: **end function**
9: **function** log_prob($h, y$)
10:    $y \leftarrow \frac{r^L}{a} y$      {Scaling to $[0, 1)$}
11:    $v_x, v_y, v_z \leftarrow \text{split}(y; 3)$      {Separating the Three Components of $y$}
12:    $h_x \leftarrow \text{SiLU}(\varphi_1(h))$
13:    $m_x \leftarrow \text{BMM1}(h_x)$
14:    $\log q \leftarrow \log\_\text{prob}(m_x, v_x)$      {Computing Log-Densities using the Built BMM Model}
15:    $h_y \leftarrow \text{SiLU}(\text{concat}(h_x, v_x)W_1)$      {Integrating the Context and the Previous Component into Features}
16:    $m_y \leftarrow \text{BMM1}(h_y)$
17:    $\log q \leftarrow \log q + \log\_\text{prob}(m_y, v_y)$
18:    $h_z \leftarrow \text{SiLU}(\text{concat}(h_y, v_y)W_2)$
19:    $m_z \leftarrow \text{BMM1}(h_z)$
20:    $\log q \leftarrow \log q + \log\_\text{prob}(m_z, v_z)$
21:    **return** $\log q + 3(L \log r - \log a)$      {Change-of-Variables Formula}
22: **end function**

---

# F. Training and Evaluation Details

## F.1. Training Details

**Two-Stage Training Scheme**      In practice, we employ a two-stage training strategy to improve performance:

1. **Stage I.** We first train the model using only the negative log-likelihood objective $\mathcal{L}_{\text{NLL}}$, with $\lambda_1 = 1$ and $\lambda_2 = 0$. The learning rate is set to 1e-3.

2. **Stage II.** After convergence of the first stage, we continue training by incorporating the energy-matching objective $\mathcal{L}_{\text{energy}}$, thereby exploiting the force-field labels to rescale and refine the energy landscape. In this stage, we set $\lambda_1 = 1$ and $\lambda_2 = 0.01$, with a learning rate of 2e-4.

Specifically, we find that jointly optimizing both objectives from scratch leads to a substantial degradation in the convergence of the log-likelihood. This is likely because the model allocates excessive attention to adjusting the relative scale of the predicted energies before it has established a reliable likelihood model. As a result, the optimization becomes unstable and struggles to capture the underlying data distribution effectively, which motivates our adoption of the two-stage training scheme. Ablation studies on the proposed two-stage training strategy are provided in Section 4.4, highlighting the critical role of the energy-alignment term in steering the learned distribution toward the true Boltzmann distribution.

**Optimizer and Scheduler**      We adopt the AdamW optimizer for model training, with momentum parameters $\beta_1 = 0.9$ and $\beta_2 = 0.95$, and a weight decay of 1e-2 (Loshchilov & Hutter, 2019). Training is performed using BF16 precision to improve computational efficiency while preserving numerical stability. We employ a learning-rate warm-up schedule consisting of 4,000 warm-up steps. Following the warm-up phase, the learning rate is controlled by a cosine-annealing scheduler with a cycle length of 100 epochs and a minimum learning rate of 1e-6 (Loshchilov & Hutter, 2017). The scheduler is updated once per epoch.

**Training Algorithm**      In summary, we present the training algorithm of CARD in Algorithm 3.

---

**Algorithm 3** Training Algorithm of CARD

---

**Require:** Training systems $\mathcal{S} = \{S_1, S_2, \cdots\}$, embedding $\mathcal{H}_\iota$, encoder $\mathcal{E}_\phi$, decoder $\mathcal{D}_\xi$, mapping to log-densities $\mathcal{F}_\psi$, batch size $B$, number of reference structures $R$

1:  Initialize all trainable parameters $\theta = \{\iota, \phi, \xi, \psi\}$
2:  **while** $\theta$ have not converged **do**
3:      Sample $S = \{U, p, z, \mathcal{C}, u\} \sim \mathcal{S}$
4:      Sample $x = \{x^{(b)}\}_{b=1}^B \sim p$                                              {Sampling Structures for a Mini-Batch}
5:      $o \leftarrow$ generate_order$(z, u, \mathcal{C})$                                    {Determining the Atom Ordering}
6:      $z, u, x \leftarrow$ reorder$(z, u, x; o)$                                            {Reordering Atoms}
7:      $u \leftarrow$ pca_alignment$(u)$, $x \leftarrow$ pca_alignment$(x)$                  {Aligning Structures to a Unique Pose}
8:      $\{s^{(b)}\} \leftarrow \{\text{to\_seq}(x^{(b)})\}_{b=1}^B$                          {Converting Structures to Sequences According to Equation (14)}
9:      $z', u', x' \leftarrow$ input_expansion$(z, u, x)$                                    {Expanding Raw Inputs to Match the Sequence}
10:     $h \leftarrow \mathcal{H}_\iota(z')$
11:     $\{h_e^{(i)}\}_{i=1}^R \leftarrow \mathcal{E}_\phi(h, u', u)$
12:     $h_e \leftarrow \frac{1}{R} \sum_{i=1}^R h_e^{(i)}$                                   {Mean Pooling over References}
13:     $\{h_d^{(b)}\}_{b=1}^B \leftarrow \mathcal{D}_\xi(h_e, x', u)$
14:     $\{\log q_\theta^{(b)}\}_{b=1}^B \leftarrow \{\text{log\_prob}(\mathcal{F}_\psi(h_d^{(b)}), s^{(b)})\}_{b=1}^B$    {Log-Density Computation}
15:     $\mathcal{L}_{\text{NLL}} \leftarrow -\frac{1}{BN} \sum_{b=1}^B \sum_{i=1}^N \log q_\theta^{(b)}[i]$
16:     $\{U_\theta^{(b)}\}_{b=1}^B \leftarrow \{-\sum_{i=1}^N \log q_\theta^{(b)}[i]\}_{b=1}^B$
17:     $\{\tilde{U}_\theta^{(b)}\}_{b=1}^B \leftarrow \{U_\theta^{(b)} - \frac{1}{B} \sum_{i=1}^B U_\theta^{(i)}\}_{b=1}^B$    {Mean Centering within the Mini-Batch}
18:     $\{\tilde{U}^{(b)}\}_{b=1}^B \leftarrow \{U(x^{(b)}) - \frac{1}{B} \sum_{i=1}^B U(x^{(i)})\}_{b=1}^B$
19:     $\mathcal{L}_{\text{energy}} \leftarrow \frac{1}{B} \sum_{b=1}^B \left| \tilde{U}_\theta^{(b)} - \tilde{U}^{(b)} \right|$
20:     $\mathcal{L} \leftarrow \lambda_1 \mathcal{L}_{\text{NLL}} + \lambda_2 \mathcal{L}_{\text{energy}}$
21:     $\theta \leftarrow$ optimizer$(\mathcal{L}; \theta)$
22: **end while**
23: **return** $\theta$

---

---

**Algorithm 4** Inference Algorithm of CARD

---

**Require:** embedding $\mathcal{H}_\iota$, encoder $\mathcal{E}_\phi$, decoder $\mathcal{D}_\xi$, mapping to log-densities $\mathcal{F}_\psi$, radix $r$, depth $L$, number of atoms $N$, atomic numbers $z$, covalent bond indices $\mathcal{C}$, reference structures $u = \{u^{(i)}\}_{i=1}^R$

1:  Initialize $x' \leftarrow [0]_{N(L+1) \times 3}$
2:  $o \leftarrow \text{generate\_order}(z, u, \mathcal{C})$                                              {Determining the Atom Ordering}
3:  $z, u \leftarrow \text{reorder}(z, u; o)$                                                     {Reordering Atoms}
4:  $u \leftarrow \text{pca\_alignment}(u)$                                         {Aligning Structures to a Unique Pose}
5:  $z', u' \leftarrow \text{input\_expansion}(z, u)$                             {Expanding Raw Inputs to Match the Sequence}
6:  $h \leftarrow \mathcal{H}_\iota(z')$
7:  $\{h_e^{(i)}\}_{i=1}^R \leftarrow \mathcal{E}_\phi(h, u', u)$
8:  $h_e \leftarrow \frac{1}{R}\sum_{i=1}^R h_e^{(i)}$                                             {Mean Pooling over References}
9:  **for** $i = 1$ **to** $NL$ **do**
10:     $h_d \leftarrow \mathcal{D}_\xi(h_e, x', u)$
11:     $q_\theta \leftarrow \text{softmax}(\mathcal{F}_\psi(h_d))$                            {Predicting Probabilities for the $r^3$ Classes}
12:     Sample $k \sim \text{Categorical}(q_\theta[i])$
13:     $k' \leftarrow \begin{bmatrix} k_x & k_y & k_z \end{bmatrix}^\top$                        {Recovering 3D Indices from $k = (k_x k_y k_z)_r$}
14:     **if** $i \leq N$ **then**
15:        $x'[i] \leftarrow a/r \cdot k' - a/2$
16:     **else**
17:        $x'[i] \leftarrow x'[i-N] + a \cdot r^{-\lceil i/N \rceil} \cdot k'$
18:     **end if**
19:  **end for**
20:  **for** $i = NL+1$ **to** $N(L+1)$ **do**
21:     $h_d \leftarrow \mathcal{D}_\xi(h_e, x', u)$
22:     Sample $\Delta x \sim \text{BMM}(\mathcal{F}_\psi(h_d)[i])$                 {Sampling Coordinates from BMM as Residuals}
23:     $x'[i] \leftarrow x'[i-N] + a/r^L \cdot \Delta x$
24:  **end for**
25:  $x \leftarrow x'[NL+1:, \dots]$
26:  $x \leftarrow \text{recover}(x; o)$                                             {Recovering the Original Atom Order}
27:  **return** $x$

---

## F.2. Hyperparameters

The hyperparameters of CARD used in our experiments are summarized in Table 4, where the best-performing values among multiple candidates are highlighted in bold.

*Table 4.* Hyperparameters of CARD used in our experiments. Among parameters with multiple candidate values, those achieving the highest evaluation performance are highlighted in **bold**.

| Radix $r$ | Depth $L$ | Scaler $a$ | Dimension $H$ | # Heads $N_d$ | # Layers $T$ | # Mixtures $K$ | # References $R$ |
|-----------|-----------|------------|---------------|---------------|--------------|----------------|------------------|
| $[3, \mathbf{4}, 5]$ | $[2, \mathbf{3}, 4]$ | 30 Å | 512 | 8 | 8 | 16 | 10 |

## F.3. Multistate Equilibrium Free Energy Simulation Protocols

Reference free energy values for the solvation (Section 4.1) and endstate correction (Section 4.2) tasks were obtained using Multistate Equilibrium Free Energy Simulations (MFES), an alchemical FEP method that provides statistically optimal estimates of the free energy changes along the defined alchemical pathways. Our implementation of MFES follows the protocol provided by Tkaczyk et al. (2024). Specifically, we define 11 intermediate states (including the two endstates) by linearly interpolating the potential energy between the initial and the target systems. For each intermediate state, equilibrium trajectories were generated through 5 ns MD simulations, with conformations recorded every 1 ps. To prevent underestimation of the variance, the trajectories were further subsampled at 5 ps intervals to reduce temporal correlations. The free energy differences between the two endstates were then computed from these decorrelated conformations using the

MBAR estimator (Shirts & Chodera, 2008).

# G. Additional Experimental Results

## G.1. Ablation Study

To systematically evaluate the effectiveness and robustness of our proposed method, we conducted additional ablation studies on the vacuum-to-toluene free energy task.

**Atom Ordering** We first investigated the impact of different atom ordering strategies on model performance by comparing our proposed topology-guided and distance-based methods against a random ordering baseline. As shown in Table 5, both the topology-guided and distance-based strategies significantly outperform the random baseline, with the topology-guided approach yielding slightly better results. This confirms that a structurally meaningful atom ordering is vital for the autoregressive model to effectively leverage contextual information.

**Model Architecture** We subsequently evaluated the necessity of CARD's key components and its coarse-to-fine architecture. As indicated in Table 5, the full model surpasses an ablated version lacking the geometry attention bias, confirming that incorporating spatial priors from reference structures enhances performance. Furthermore, we evaluated CARD against a vanilla continuous autoregressive baseline, which is constructed by disabling radix decomposition ($L = 0$) and replacing the BMM with a Gaussian Mixture Model (GMM) to handle unbounded coordinates. The substantial margin by which CARD beats this baseline clearly demonstrates the superiority of our coarse-to-fine modeling framework.

*Table 5.* Ablation study on the effect of model architecture on performance, evaluated on the solvation free energy task from vacuum to toluene. All settings are evaluated with Stage I training only. The best result for each metric is shown **bold**.

| Model Architecture | MAE | RMSE | $R^2$ | Pct ($<1$) |
|---|---|---|---|---|
| full (topology-guided atom ordering) | **0.81** | **1.34** | **0.91** | **77.1** |
| w/ distance-based atom ordering | 1.12 | 1.67 | 0.86 | 67.1 |
| w/ random atom ordering | 1.73 | 2.50 | 0.72 | 38.0 |
| w/o decomposition (vanilla autoregressive) | 1.29 | 1.64 | 0.87 | 38.6 |
| w/o geometry attention bias | 1.17 | 1.76 | 0.86 | 64.3 |

**Reference Structures** Next, we assessed the robustness of CARD with respect to the quantity and quality of the reference structures. To evaluate quantity, we selected subsets of $\{1, 2, 4, 8\}$ reference structures from the full 10 ns MD trajectories. To evaluate quality, we sampled 10 reference structures exclusively from the initial 1 ns and 100 ps segments of the trajectories. All evaluations were performed at test time using the full model trained through both Stage I and Stage II. As detailed in Table 6, performance remains remarkably stable provided that more than one reference structure is used, demonstrating CARD's strong robustness to variations in both reference quantity and quality.

*Table 6.* Test-time ablation study on the number and quality of reference structures, evaluated on the vacuum-to-toluene solvation free energy task with the model trained on both Stage I and Stage II. The column *Reference Source* indicates the segment of the trajectory from which the reference structures were sampled. Best results are shown in **bold**.

| # References $R$ | Reference Source | MAE | RMSE | $R^2$ | Pct ($<1$) |
|---|---|---|---|---|---|
| 10 | 10 ns MD | 0.71 | 1.27 | **0.92** | 82.9 |
| 1 | | 1.02 | 2.19 | 0.76 | 71.4 |
| 2 | 10 ns MD | 0.78 | 1.39 | 0.90 | 81.4 |
| 4 | | 0.70 | 1.31 | 0.91 | **85.7** |
| 8 | | 0.78 | 1.37 | 0.91 | 80.0 |
| 10 | 1 ns MD | **0.69** | **1.25** | **0.92** | 80.0 |
| 10 | 100 ps MD | 0.79 | 1.53 | 0.89 | 80.0 |

**Training Set Size** Finally, to evaluate the model's sensitivity to training data volume, we trained the full two-stage CARD model on a randomly reduced subset of 10,000 molecules (down from the original 40,303). As shown in Table 7, limiting the training data causes only a marginal performance drop. Crucially, the MAE stays well below the 1 kcal/mol threshold for chemical accuracy, demonstrating CARD's strong data efficiency and robustness.

*Table 7.* Ablation study on the training set size, evaluated on the vacuum-to-toluene solvation free energy task with the model trained on both Stage I and Stage II. Best results are shown in **bold**.

| # Training Molecules | MAE | RMSE | $R^2$ | Pct ($<1$) |
|---|---|---|---|---|
| 40,303 | **0.71** | **1.27** | **0.92** | **82.9** |
| 10,000 | 0.84 | 1.44 | 0.90 | 76.5 |

## G.2. Scalability to Peptides

To demonstrate the scalability of CARD to larger systems, we evaluated its performance on peptides by comparing it against two strong Boltzmann generators: Prose (Tan et al., 2025a) and TarFlow (Zhai et al., 2025) (with the latter adapted as a Boltzmann generator in Tan et al. (2025a)). Both baselines are known to generalize across varying peptide sequence lengths. For a fair comparison, we retrained CARD on peptides up to 4 amino acids (AA) from the many-peptides-md dataset (Tan et al., 2025a), strictly following the data splits from Prose. Specifically, since the dataset does not provide ground-truth energy labels, the model was trained exclusively via Stage I without the energy-alignment term.

Evaluations were carried out on the 2AA and 4AA test sets, each comprising 30 peptide systems. Following the protocol in Prose, we utilized the Wasserstein-2 distance between the generated and reference MD ensembles for evaluation. This was computed on both the torus and time-lagged independent components, denoted as Torus-$\mathcal{W}_2$ and TICA-$\mathcal{W}_2$, respectively. As shown in Table 8, even when limited to Stage I training, CARD achieves performance comparable to these state-of-the-art Boltzmann generators on both the 2AA and 4AA test sets. Furthermore, it experiences noticeably less performance degradation than the baselines when transitioning from 2AA to 4AA. This demonstrates not only its robust scalability to larger systems of up to 100 atoms, but also its superior capacity for sequence length generalization.

*Table 8.* Unweighted proposal performance on the 2AA and 4AA test sets of the many-peptides-md dataset. Each method generates 10,000 proposals for evaluation without resampling. $\mathcal{W}_2$ denotes the Wasserstein-2 distance. The best result for each metric is **bolded**.

| Model | 2AA (30 systems) | 4AA (30 systems) | |
|---|---|---|---|
| | Torus-$\mathcal{W}_2$ ($\downarrow$) | Torus-$\mathcal{W}_2$ ($\downarrow$) | TICA-$\mathcal{W}_2$ ($\downarrow$) |
| TarFlow | **0.178** | 0.882 | **0.384** |
| Prose | 0.261 | 0.916 | 0.546 |
| CARD (Stage I) | 0.296 | **0.832** | **0.384** |

Furthermore, we compared the computational efficiency of CARD against Prose in Table 9. While CARD achieves comparable performance with significantly fewer parameters, its training throughput and inference memory footprint remain on the same scale as Prose. This suggests that although CARD is highly scalable and parameter-efficient, its current implementation and underlying architecture still have room for system-level engineering optimization.

*Table 9.* Model size and computational cost comparison between Prose and CARD on the many-peptides-md dataset. CARD is trained on sequences up to length 4, while Prose is trained on sequences up to length 8. Peak GPU memory is measured during inference on the same device and reported in mean±std, using a batch size of 512 per test peptide.

| Model | # Parameters (M) | Training Iterations | Peak GPU Memory Allocated (GB) | |
|---|---|---|---|---|
| | | | 2AA (30 systems) | 4AA (30 systems) |
| Prose | 285 | 260 H100 hours | 3.33±0.19 | 4.21±0.27 |
| CARD | 44.5 | 640 A100 hours | 2.30±0.53 | 5.35±1.18 |

### G.3. Overlap Diagnostic

We conducted overlap diagnostics on the vacuum-to-toluene solvation free energy task to determine the reliability of the predicted values. For each test system, we computed the bidirectional reweighting effective sample sizes (ESS) between the CARD proposal (state 0) and the reference MD ensembles (state 1) in both vacuum and toluene environments. For both the $0 \rightarrow 1$ and $1 \rightarrow 0$ directions, we recorded the minimum ESS across the vacuum and toluene phases to serve as a conservative lower bound for the phase-space overlap. As shown in Figure 8, we plotted the absolute error of the predicted relative free energies (with respect to MFES) against the minimum, maximum, and harmonic mean of $ESS_{0 \rightarrow 1}$ and $ESS_{1 \rightarrow 0}$. Notably, both the maximum and the harmonic mean value of the bidirectional ESS exhibit strong Spearman correlations, demonstrating their utility as practical indicators of potential failure modes.

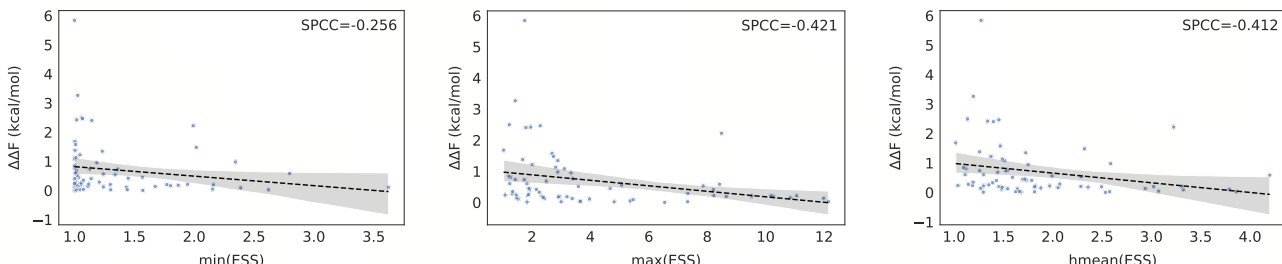

*Figure 8.* Correlation between free energy prediction error and the overlap of CARD proposal (state 0) and MD ensembles (state 1) in the vacuum-to-toluene solvation free energy task. Overlap is quantified using Effective Sample Size (ESS), computed in both directions ($0 \rightarrow 1$ and $1 \rightarrow 0$). For each molecule, the minimum (left), maximum (middle), and harmonic mean (right) of its two directional ESS values are used as the x-axis to plot the panels.

### G.4. PCA Alignment Instability Analysis

We assessed the potential instability of PCA alignment using the 3D conformers provided in the SDF files. For each molecule, we performed PCA on the centered atomic coordinates and checked whether any pair of principal axes was nearly degenerate (*i.e.*, having a relative difference in eigenvalues within a fixed tolerance of 0.02). Molecules meeting this near-degeneracy criterion were flagged as potential alignment failures, and the overall failure rate was calculated as the fraction of flagged cases across the dataset. The resulting failure rate was remarkably low, accounting for only 0.057% (23/40,403) of the entire ZINC-derived dataset, and dropping to exactly 0% across all test sets. This confirms that PCA degeneracy is highly uncommon among these small molecules and does not compromise the validity of our results.

## H. Computational Infrastructure

As described in Section F.1, we employ a two-stage training strategy for our model. Stage I training was performed on 32 NVIDIA A100 SXM 80GB GPUs with a batch size of 200 and a maximum of 100 epochs, with a single model converging in approximately 4 days (around 40 epochs). Stage II training was conducted on 16 NVIDIA A100 SXM 80GB GPUs using the same batch size and epoch limit, with each model requiring roughly 10 days to complete. All inference tasks were carried out on NVIDIA Tesla V100 SXM2 32GB GPUs.

