# OpenReview forum: "CARD: Coarse-to-fine Autoregressive Modeling with Radix-based Decomposition for Transferable Free Energy Estimation"
_ICML.cc/2026/Conference — ICML 2026 regular_

### Official Review · Reviewer_kmwG · 2026-02-27

**Soundness:** 3
**Presentation:** 3
**Significance:** 3
**Originality:** 3
**Overall Recommendation:** 4
**Confidence:** 4

**Summary:**

This paper proposes CARD (Coarse-to-fine Autoregressive Modeling with Radix-based Decomposition), a transformer-based generative framework for transferable free energy estimation. The method introduces a radix-based decomposition that bijectively maps 3D molecular coordinates into mixed discrete-continuous sequences, enabling coarse-to-fine autoregressive modeling. A central concept presented by the study is the use of tractable generative models as zero-free-energy references, allowing absolute free energy computation via BAR without alchemical intermediates. The authors claim to assess a central concept in deep-learning-based free energy estimation: whether expressive autoregressive models can generalize across molecular systems with different topologies while preserving tractable likelihoods. Empirical evaluations on solvation free energy, endstate correction, and aqueous tautomer free energy show promising accuracy and reported inference speedups.

**Compliance With Llm Reviewing Policy:**

Affirmed.

**Final Justification:**

The authors addressed some of the issues I raised in the original review.

**Key Questions For Authors:**

1. How does CARD compare empirically against diffusion-based conformational generative models or Boltzmann Generators trained under identical settings?

2. What is the parameter count and memory footprint relative to flow-based or diffusion-based baselines?

3. Can the model be made fully SE(3)-equivariant instead of relying on PCA alignment?

4. How does performance scale with molecular size (e.g., 50-200 atoms)?

5. Why are multiresolution and hierarchical graph modeling works not cited or compared, given the conceptual overlap?

6. Is the 40x speedup still significant when accounting for training time?

**Limitations:**

While theoretically elegant, the method relies on PCA alignment to remove roto-translational degrees of freedom, which is not fully symmetry-consistent and may introduce instability for symmetric or near-symmetric structures. The radix decomposition increases sequence length by a factor of L, which may negatively affect computational scaling for large systems. Furthermore, the framework assumes access to MD trajectories for training under each thermodynamic condition, limiting its practicality in data-scarce regimes. Most importantly, the empirical validation lacks comparison with state-of-the-art generative molecular models and multiresolution graph-based approaches, making it difficult to assess whether the proposed contribution is fundamentally superior or mainly incremental in practice. For an ICML submission, the experimental depth and comparative rigor are currently insufficient.

**Strengths And Weaknesses:**

*** Strengths: The paper is technically ambitious and grounded in statistical mechanics. The theoretical connection between tractable densities and zero free energy references is clearly articulated and builds upon prior work such as DeepBAR. The radix-based decomposition is novel and enables exact likelihood computation in a nontrivial coordinate-generation setting. The model shows encouraging transferability across unseen molecular systems, which is an important practical goal. The reported 40-fold inference speedup over classical MFES is potentially impactful for large-scale screening applications.

*** Weaknesses:

1. Missing literature on coarse-graining and multiresolution molecular modeling, especially learning-to-cluster and hierarchical approaches such as: [1], [2] and [3]. These works are directly relevant to coarse-to-fine, multiscale, and hierarchical modeling of molecular or geometric structures and should be discussed and compared conceptually.

2. Lack of strong empirical baselines. The paper compares primarily against classical MFES, DFT, and sPhysNet, but does not include:
- Modern diffusion-based generative models for molecular conformations
- Boltzmann Generators / normalizing flow approaches (beyond conceptual discussion)
- Flow matching or diffusion transport-based free energy estimators
- Multiscale generative baselines

3. No ablation against simpler autoregressive baselines (e.g., direct coordinate autoregression without radix decomposition).

4. Limited scalability analysis. Although inference time is reported, there is no detailed comparison in terms of:
- Parameter count
- Memory footprint
- Scaling behavior with atom count
- Comparison to diffusion or flow-based architectures

5. SE(3) handling is heuristic. PCA alignment introduces potential instability for symmetric molecules and breaks full equivariance; this is weaker than modern equivariant architectures. For example, please check [4].

6. Evaluation scope is narrow. Experiments focus on drug-like small molecules; no demonstration on larger systems (e.g., protein-ligand complexes).

7. Generalization claims lack stress tests. True topology extrapolation or OOD evaluation is not rigorously analyzed.

*** References

[1] Nhat Khang Ngo, Truong-Son Hy, and Risi Kondor, Multiresolution Graph Transformers and Wavelet Positional Encoding for Learning Long-Range and Hierarchical Structures, Journal of Chemical Physics, Volume 159, Issue 3, DOI 10.1063/5.0152833.
URL: https://doi.org/10.1063/5.0152833

[2] Truong-Son Hy and Risi Kondor, Multiresolution Equivariant Graph Variational Autoencoder, Machine Learning: Science and Technology, Volume 4, Number 1, DOI 10.1088/2632-2153/acc0d8.
URL: https://iopscience.iop.org/article/10.1088/2632-2153/acc0d8

[3] Thuan Nguyen Anh Trang, Khang Nhat Ngo, Hugo Sonnery, Thieu Vo, Siamak Ravanbakhsh, Truong-Son Hy, Scalable Hierarchical Self-Attention with Learnable Hierarchy for Long-Range Interactions, Transactions on Machine Learning Research (TMLR).
URL: https://openreview.net/pdf?id=qH4YFMyhce

[4] Brandon Anderson, Truong-Son Hy and Risi Kondor, Cormorant: Covariant molecular neural networks, NeurIPS 2019.
URL: https://dl.acm.org/doi/10.5555/3454287.3455589

---

> ### Author Rebuttal · Authors · 2026-03-31
>
> We appreciate the reviewer's insightful suggestions and helpful feedback, and we address the questions and concerns in detail below.
>
> >W1&Q5
>
> **A1**:
> We thank the reviewer for the helpful suggestion. We have expanded the *Related Work* section to include a discussion of [1]–[3] and related multiscale molecular modeling methods. In brief, prior works focus on multiresolution graph architectures that learn molecular substructures to improve attention efficiency and scalability. In contrast, CARD performs multilevel modeling directly on all-atom 3D coordinates to address a key challenge in autoregressive generation—committing to local coordinates before global geometry is established—while preserving exact likelihood evaluation through a unique coordinate-level decomposition. This distinguishes CARD from prior hierarchical graph-based approaches.
>
> >W2&W6&Q1&Q4
>
> **A2**: We thank the reviewer for the suggestion. We supplement our study with experiments on the many-peptides-md dataset [5]. We compare CARD with Prose [5] and TarFlow [6], where TarFlow is adapted for this task as a Boltzmann generator in [5]; both models generalize across peptide sequence lengths. Due to space limitations, we kindly refer the reviewer to our response to Reviewer s8GM (**A1**) for details.
>
> >W3
>
> **A3**:
> We thank the reviewer for this helpful suggestion. We have added a simpler autoregressive baseline based on direct coordinate autoregression by disabling radix decomposition ($L=0$) and replacing the BMM with a Gaussian Mixture Model (GMM) since the coordinates are unbounded, while keeping other hyperparameters the same. Results in [Table A](https://anonymous.4open.science/r/CARD-88DE/materials/ablation-arc.png) show that CARD consistently outperforms the vanilla autoregressive model on the vacuum-to-toluene solvation free energy task, showing the superiority of the coarse-to-fine framework.
>
> Moreover, if interested, please refer to our response to Reviewer YsD4 (**A7**) for additional ablation studies.
>
> >W4&Q2
>
> **A4**: Thank you for the suggestion. We compare CARD with Prose [5], a Boltzmann generator generalizable to peptide sequence lengths, on the many-peptides-md dataset [5] in terms of parameter count, training throughput, and inference memory footprint. As shown in [Table B](https://anonymous.4open.science/r/CARD-88DE/materials/cost.png), CARD uses fewer parameters, while training cost and inference memory footprint remain on a similar scale, likely due to differences in architecture and training dynamics.
>
> >W5&Q3: SE(3)-equivariance.
>
> **A5**:
> Thank you for this important question. While SE(3)-equivariant models are a natural choice in principle, they are not directly applicable here: obtaining a tractable likelihood requires fixing the global SE(3) gauge to define the conditional distributions, which is unstable when the molecular structure is only partially generated. We therefore adopt PCA alignment as a practical gauge-fixing step.
>
> To address concerns about PCA instability for symmetric molecules, we quantified the failure rate by checking for near-degenerate principal axes. The failure rate is extremely low: 0.057% (23/40,303) on the training set and 0% on all the test sets. This indicates that PCA instability is rare for these small molecules and does not impact our practical results.
>
> >W7: Generalization claims lack stress tests.
>
> **A6**: Thank you for the comment. To ensure that the test sets are chemically distinct from the training sets, we performed a Tanimoto similarity analysis using ECFP4 fingerprints. We filtered the Section 4.1 test set to retain 70 distinct cases (all cases in Section 4.2 already satisfy this criterion). Due to space limitations, detailed similarity metrics and updated results on the filtered set are provided in our response to Reviewer YsD4 (Reply **A1**).
>
> >Q6
>
> **A7**: Thank you for the question. While training time is indeed a factor, the 40× inference speedup over physics-based methods remains practically significant. Because CARD generalizes to unseen molecules, its training cost is a one-time investment. Unlike prior methods (e.g., FEAT [4]) that require retraining for each new system, CARD can be directly applied to new molecules, making inference speed the dominant factor in practical large-scale efficiency bottleneck.
>
> **References**
>
> [1] Ngo, N. K. et al. (2023). Multiresolution Graph Transformers and Wavelet Positional Encoding for Learning Long-Range and Hierarchical Structures.
>
> [2] Hy, T.-S. et al. (2023). Multiresolution Equivariant Graph Variational Autoencoder.
>
> [3] Nguyen Anh Trang, T. et al. (2024). Scalable Hierarchical Self-Attention with Learnable Hierarchy for Long-Range Interactions.
>
> [4] Du, Y. et al. (2025). FEAT: Free energy Estimators with Adaptive Transport.
>
> [5] Tan, C. B. et al. (2025). Amortized Sampling with Transferable Normalizing Flows.
>
> [6] Zhai, S. et al. (2024). Normalizing flows are capable generative models.

---

> > ### Author Rebuttal · Reviewer_kmwG · 2026-04-05
> >
> > The authors have addressed some of the issues I raised in the original review. I increase the rating to Weak Accept.
> >
> > However, I respectfully disagree with the authors. In physics, symmetry is important. Thus, I think the authors should find a way to make their physical models symmetry-preserving.

---

> > > ### Author Response · Authors · 2026-04-05
> > >
> > > Dear Reviewer kmwG,
> > >
> > > Thank you for the positive feedback and for increasing the score. We would like to further elaborate on the reviewer's insightful point regarding equivariance.
> > >
> > > First, we fully concur that SE(3)-equivariance is fundamentally critical for physical modeling. The absence of intrinsic equivariance in CARD is a deliberate engineering trade-off made to seamlessly accommodate our autoregressive generative framework. To compensate for the lack of intrinsic equivariance, we implemented a PCA-based alignment procedure as a practical alternative. Empirically, this approach has proven to be robust, yielding a failure rate of just 0.057% on the training set and 0% on all the test sets.
> > >
> > > Second, we wish to emphasize that the lack of strictly hardcoded equivariance does not compromise the model's reliability or predictive power. This design philosophy is well-supported by the breakthroughs in the field. For instance, AlphaFold3 [1] deliberately abandons strict structural equivariance (e.g., invariant point attention) in its diffusion module. Instead, it relies on extensive data augmentation and large-scale training to implicitly learn symmetry-related behaviors from data. CARD operates on a similar premise, prioritizing architectural flexibility and the critical ability to estimate exact likelihoods over explicit symmetry constraints.
> > >
> > > That being said, we deeply appreciate the reviewer's suggestion and consider it highly constructive. Developing a principled methodology to natively integrate SE(3)-equivariance into an autoregressive framework without sacrificing its inherent efficiency is is an active avenue we plan to explore in our future work.
> > >
> > > Once again, we sincerely thank you for your constructive engagement throughout the review process, which has been instrumental in refining the clarity and quality of our manuscript.
> > >
> > > **Reference**
> > >
> > > >[1] Abramson, J., Adler, J., Dunger, J., Evans, R., Green, T., Pritzel, A., ... & Jumper, J. M. (2024). Accurate structure prediction of biomolecular interactions with AlphaFold 3. Nature, 630(8016), 493-500.

---

### Official Review · Reviewer_s8GM · 2026-03-12

**Soundness:** 2
**Presentation:** 3
**Significance:** 3
**Originality:** 3
**Overall Recommendation:** 3
**Confidence:** 3

**Summary:**

The paper discusses the problem of estimating free energy differences in molecular systems. The main idea is to learn a tractable autoregressive generative model over molecular conformations and use that as a zero-free-energy proposal, so that absolute free energies can be computed with BAR (Bennett Acceptance Ratio). The authors also used a PCA-based canonical frame to align molecules coordinates , then convert the coordinates into a mixed sequence through a radix-based decomposition, which is modeled by a Transformer backbone (encoder-decoder style, where the encoder processes reference structures and the decoder predicts the probability of the conformations). The authors evaluated their method on three tasks:  Solvation free energy prediction, Endstate Correction, and aqueous tautomer free energy estimation.

**Compliance With Llm Reviewing Policy:**

Affirmed.

**Final Justification:**

The rebuttal addressed some of my concerns and I raised my score. I think the paper should provide more study if the method is  applicable to other domains.

**Key Questions For Authors:**

- For training cost, the authors stated that stage 1 training takes around 4 days on 32 A100 GPUs, and stage 2 takes an additional 10 days on 16 A100 GPUs. How this compare to other deep learning methods?

- I understand the framework based on the learned distribution $q_\theta$ being a good approximation of the true Boltzmann distribution $p$. Since the model is trained to maximize log-likelihood, does this metric guarantee that all conformational modes are captured?

**Limitations:**

Yes.

**Strengths And Weaknesses:**

**Strengths:**

* The authors evaluate their method on three different tasks. They also show strong empirical results and clear improvements over existing baselines on aqueous tautomer free energy estimation, outperforming both DFT and sPhysNet.

* The proposed framework achieves faster inference compared to classical computational methods (MFES pipelines), as it needs to sample from the two endpoint distributions.

**Weaknesses:**

* The empirical comparison is still limited. The authors should include more baselines and compare their framework against learned free-energy methods.

*  Some evaluations rely on relatively small test sets. The endstate correction evaluation uses only 18 molecules from HiPen, and the tautomer task uses 27 pairs after filtering.

* Limited to small drug-like molecules. All evaluations are on small organic molecules. The authors acknowledge that extension to protein-ligand complexes could be a futrue work, but this might limits the strengths of the paper.

---

> ### Author Rebuttal · Authors · 2026-03-31
>
> We thank the reviewer for the thoughtful and detailed comments and address the questions below.
>
> >W1: The empirical comparison is still limited.
>
> **A1**:
> We thank the reviewer for the suggestion regarding baselines. Firstly, we would like to clarify that our primary focus is **exact free-energy estimation** rather than conformation sampling alone. As described in Section 2.2, our estimator treats the learned distribution as a reference state with zero absolute free energy, which requires a generative model with tractable exact likelihood (Eq. 7). Many modern diffusion or flow-matching models, while effective samplers, do not support efficient exact likelihood computation and are therefore not appropriate to compare with CARD in this setting.
>
> Among models that support exact likelihood or free-energy computation, many methods (e.g., Transferable Boltzmann Generators [1] and FEAT [2]) require fixed-dimensional inputs and are trained per system, and thus do not generalize to unseen systems. We therefore compare with Prose [3] and TarFlow [7], with TarFlow adapted for this task as a Boltzmann generator in [3], both of which generalize across peptide sequence lengths. For a fair comparison, we retrain and evaluate CARD on the many-peptides-md dataset using the same split, training on peptides up to 4 AA and evaluating on the 2AA and 4AA test sets. Results in [Table A](https://anonymous.4open.science/r/CARD-88DE/materials/pep.png) show that CARD achieves performance comparable to state-of-the-art Boltzmann generators, even when trained using only Stage I (without the energy-alignment term).
>
> >W2: Some evaluations rely on relatively small test sets.
>
> **A2**:
> We thank the reviewer for raising this point. While the HiPen end-state correction benchmark contains only 18 molecules, it is a curated stress-test set selected in prior work to include particularly challenging cases for CHARMM-style parameterization, and even one-step FEP can exhibit substantial deviations from reference values on these systems [4]. Therefore, achieving accurate corrections on HiPen is non-trivial and practically meaningful despite the modest size.
>
> For the tautomer benchmark, we follow the data split used in prior work [5] to enable a direct, fair comparison. Our additional filtering removes tautomer pairs that are effectively single-state at ambient conditions, which are of limited practical interest and can otherwise bias the evaluation.
>
> >W3: Limited to small drug-like molecules.
>
> **A3**: We thank the reviewer for this comment. We agree that extending CARD to protein–ligand complexes would be an important next step; however, doing so will likely require additional engineering and architectural optimizations to scale to much larger, heterogeneous systems. As a proxy, we train and evaluate the model on the many-peptides-md dataset, a multi-peptide benchmark, which demonstrates its scalability with respect to molecular size (see **A1**). We believe these results already represent an important step forward in generalizable free-energy estimation.
>
> >Q1: How do training throughputs compare to other deep learning methods?
>
> **A4**: We thank the reviewer for the question on training cost. We have added a comparison with Prose [3] on the many-peptides-md dataset, including parameter count, training throughput, and memory footprint. Due to space limits, details are provided in our response to Reviewer kmwG (Reply **A4**).
>
> >Q2: Since the model is trained to maximize log-likelihood, does this metric guarantee that all conformational modes are captured?
>
> **A5**: Thank you for the question. In the idealized setting, maximizing expected log-likelihood recovers the full target distribution, including all modes, under standard regularity conditions [6]. In practice, rare conformations may be absent from the training set due to limited MD sampling or dataset imbalance. We argue this is acceptable for two reasons. First, CARD is trained as a generalizable model and can capture such transitions from large-scale MD data across other systems. Second, truly rare conformations have low Boltzmann probability and thus contribute little to the free-energy integral, so their absence has minimal impact on the final estimate. Together, these points justify the use of maximum log-likelihood training in our framework.
>
> **References**
>
> [1] Klein, L. et al. (2024). Transferable boltzmann generators.
>
> [2] Du, Y. et al. (2025). FEAT: Free energy Estimators with Adaptive Transport.
>
> [3] Tan, C. B. et al. (2025). Amortized Sampling with Transferable Normalizing Flows.
>
> [4] Tkaczyk, S. et al. (2024). Reweighting from Molecular Mechanics Force Fields to the ANI-2x Neural Network Potential.
>
> [5] Pan, X. et al. (2025). Fast and accurate prediction of tautomer ratios in aqueous solution via a siamese neural network.
>
> [6] Murphy, K. P. (2022). Probabilistic machine learning: an introduction.
>
> [7] Zhai, S. et al. (2024). Normalizing flows are capable generative models.

---

> > ### Author Rebuttal · Reviewer_s8GM · 2026-04-03
> >
> > I thank the authors for their responses and the efforts in the rebuttal, which have addressed some of my concerns. I appreciate the clarifications on the evaluation test. I'll update my score accordingly. However, I think there are still some remaining points in the applicability of the method to other domains.

---

> > > ### Author Response · Authors · 2026-04-03
> > >
> > > Dear Reviewer s8GM,
> > >
> > > We thank the reviewer for the positive feedback. Regarding the applicability of CARD to broader domains, we would like to clarify the following three aspects:
> > >
> > > First, theoretically, CARD is a domain-agnostic framework. As an all-atom generative model, it relies solely on fundamental physical inputs at inference time, namely, atom types, molecular topology (i.e., covalent bonds), and a few reference structures sampled from MD trajectories. By avoiding domain-specific features or heuristics, CARD is fundamentally generalized and can be applied to a wide spectrum of molecular systems.
> > >
> > > Second, empirically, CARD has demonstrated robust generalization across diverse scenarios. In our original manuscript, we extensively evaluated CARD on small molecules across multiple tasks, including solvation free energies in different solvents, endstate corrections across force fields, and tautomer free energy calculations. These results highlight both the versatility and the substantial computational efficiency of CARD over traditional physics-based methods (e.g., MFES). Furthermore, following the reviewer's constructive suggestion, we have added new experiments (Reply **A1**) on the many-peptides-md dataset during the rebuttal period. Impressively, even without the energy-alignment term during training, CARD achieves performance comparable to state-of-the-art Boltzmann Generators (e.g., Prose), firmly validating its practical applicability to larger peptide systems (**up to over a hundred atoms**).
> > >
> > > Finally, scaling to macroscopic systems is an engineering hurdle rather than a methodological limitation. We fully agree that extending CARD to large protein–ligand complexes is a highly valuable direction. Currently, our implementation utilizes a standard attention architecture without incorporating memory-efficient optimizations (e.g., FlashAttention). As a result, there may be efficiency challenges when scaling to systems with thousands of atoms, which are prohibitive to resolve within the short rebuttal window. That said, we believe this is primarily an implementation and engineering issue rather than a limitation of the CARD framework itself. Integrating efficient attention mechanisms and adapting the coarse-to-fine modeling paradigm to efficiently scale CARD for large biomolecular complexes (thousands of atoms) will be a prioritized focus of our future work.
> > >
> > > We hope these clarifications address your concerns, and we sincerely appreciate your constructive engagement, which has significantly helped us refine the positioning of our work.

---

### Official Review · Reviewer_YsD4 · 2026-03-12

**Soundness:** 4
**Presentation:** 3
**Significance:** 3
**Originality:** 3
**Overall Recommendation:** 5
**Confidence:** 4

**Summary:**

This paper proposes CARD, a transformer-based autoregressive generative model for estimating absolute free energies of molecular systems. The core idea builds on the zero-free-energy proposal paradigm (DeepBAR): any normalized probability density $q_\theta$, when interpreted as a Boltzmann distribution with energy $U_\theta = -\log q_\theta$, has partition function $Z_\theta = 1$ and hence free energy $F_\theta = 0$. This allows the Bennett Acceptance Ratio (BAR) to be used between the model and a physical system to directly recover the absolute free energy of the physical system, bypassing alchemical intermediate states. CARD's main technical contribution is a radix-based decomposition that bijectively converts 3D coordinates into mixed discrete-continuous sequences, enabling coarse-to-fine autoregressive generation with exact, tractable log-likelihoods. The model is conditioned on system context (atomic numbers, bond topology, reference structures), allowing generalization to unseen molecules without retraining. Experiments on solvation free energies, endstate corrections (MM→NNP), and aqueous tautomer free energies show accuracy comparable to classical multistate equilibrium free energy simulations (MFES).

**Compliance With Llm Reviewing Policy:**

Affirmed.

**Final Justification:**

This work is a meaningful advance of the field of ML-estimation of free energy differences by demonstrating transferability of those methods for the first time. The authors have addressed all concerns well, improving the evaluation and the rigor of the provided results. I have improved my overall score, reflecting the strengthened evaluation and the added ablations. I trust the authors will improve the structure/readability of the paper upon acceptance.

**Key Questions For Authors:**

1. In example 4.3, did the authors attempt to model the change in free energy by directly taking the difference in absolute free energy between both tautomers in water without the thermodynamic cycle in vacuum? Would it significantly reduce the accuracy when taking the difference in absolute free energy in implicit water with the MM force field rather than leveraging ANI2x via the thermodynamic cycle?

**Limitations:**

The authors discuss PCA instability and restriction to drug-like molecules. Additional limitations include:
- implicit solvent and neutral-only molecules limit practical applicability
- the 40× speedup excludes endpoint MD costs, potentially reducing to 3–5× realistically
- sharp hyperparameter sensitivity suggests limited transferability to new domains
- accuracy is limited to the accuracy of the underlying MD simulation

**Strengths And Weaknesses:**

### Strengths

**S1. Transferability across molecular systems.** This is a meaningful practical advance over DeepBAR and other theory-based DL methods for free energy estimation. By conditioning on system context $c = (\{z, \mathcal{C}, u\}$) and using a variable-length autoregressive architecture, a single trained model serves as the zero-free-energy reference for arbitrary molecules without retraining, a limitation that largely negated the speed advantage of prior deep learning approaches.

**S2. Principled theoretical framework.** The zero-free-energy proposal idea is elegant: any normalized density has $F_\theta = 0$ and can anchor BAR for absolute free energy computation. The choice of autoregressive modeling over normalizing flows (DeepBAR) or diffusion models is well-justified - autoregressive models provide exact log-likelihoods, whereas diffusion models cannot efficiently compute the densities required for the $F_\theta = 0$ property.

**S3. Creative radix decomposition.** Converting continuous 3D coordinates into a mixed discrete-continuous sequence ordered by resolution level rather than by atom is a genuinely novel idea. It lets the model establish coarse positions of all atoms before refining any of them - a natural inductive bias for molecular structures.

**S4. Strong solvation results and encouraging tautomer experiment.** Sub-1 kcal/mol MAE on solvation free energies with $R^2 > 0.9$ is encouraging for a generalizable model. The zero-shot tautomer experiment is conceptually compelling - composing independently trained models without additional training demonstrates the modularity of absolute free energy computation. Figure 5 is particularly informative: CARD's errors remain stable as molecular complexity increases (measured by rotatable bonds), while DFT deteriorates with larger outliers.

**S5. Thoughtful two-stage training.** The observation that joint NLL + energy alignment destabilizes convergence, and the fix of first learning distribution shape then refining energy scale, is a practical insight well-supported by ablation evidence (Table 4).

### Weaknesses

**W1. Random train/test split may overestimate generalization.** The 40,303 ZINC20 molecules are randomly split with only 100 held out for testing. No scaffold-based or similarity-based splitting is used, so chemically similar molecules likely appear on both sides. A scaffold split or Tanimoto similarity analysis would substantially strengthen the generalization claims. The same concern applies to the endstate correction task (18 HiPen test molecules vs 7,881 ZINC20 training molecules).

**W2. The 40× speedup excludes significant costs.** MFES's approx. 32,000s per molecule is self-contained. CARD's ~800s excludes: (a) endpoint MD trajectories for new molecules (10 ns per endpoint, potentially adding thousands of seconds); (b) the $R=10$ reference structures that must come from MD. The paper should report the full per-molecule cost for a genuinely new molecule or provide a break-even analysis. The amortized training cost (approx. 18,000 GPU-hours across three models, ~1.2M ns of training MD) should also be acknowledged.

**W3. Evaluation uses MFES as ground truth rather than experiment.** The 0.78 kcal/mol MAE for solvation measures agreement between two computational methods sharing the same GAFF force field, not physical accuracy. Experimental solvation free energies are available (e.g., FreeSolv). Comparing both CARD and MFES against experiment would disentangle method validation (does CARD faithfully recover the force field's free energies?) from utility validation (are those free energies meaningful?).

**W4. Overlap diagnostics and failure mode analysis are absent.** The framework's reliability hinges on BAR overlap between $q_\theta$ and the physical distribution, yet no overlap statistics, convergence diagnostics, or per-system uncertainty estimates are reported. Reporting pymbar effective sample sizes or overlap matrices would help practitioners know when to trust CARD's estimates.

**W5. SE(3) invariance relies on fragile PCA preprocessing.** Rather than building SE(3) invariance into the architecture, CARD uses PCA alignment, which fails when principal axes are near-degenerate - not only for globally symmetric molecules but for any molecule with approximate local symmetry (e.g., para-substituted rings). When alignment is unstable, similar conformations map to very different coordinates, creating discontinuities in $q_\theta$ that propagate into BAR. The paper acknowledges this but does not quantify how often it occurs in the test set.

**W6. Limited evaluation scope - implicit solvent, small neutral molecules.** All experiments use implicit solvent (GB-OBC2), and the dataset is restricted to electrically neutral molecules. Extension to explicit-solvent protein-ligand systems, where free energy estimation has the highest practical value, remains open. This is admittedly a limitation shared with competing DL-based methods, but it tempers the practical significance of the results.

**W7. Incomplete ablations.** The ablations cover radix $r$, depth $L$, and two-stage training, but leave important questions unanswered: no ablation on the number of reference structures $R$ or the quality of those references (short vs. long MD), no ablation on the geometry-aware attention mechanism (Eq. 21–22), no ablation on atom ordering strategy (topology-guided vs distance-based vs random), and no ablation on training set size.

**W8. Hyperparameter sensitivity.** Table 4 reveals sharp performance cliffs: $L=2 \to L=3$ takes $R^2$ from $-0.14$ to $0.87$; $r=3 \to r=4$ cuts MAE from 3.05 to 0.97. These are catastrophic failures at nearby settings, not gradual degradations. For new domains with different spatial scales (e.g., protein-ligand binding), the optimal $(r, L, a)$ could shift with no principled selection method.

**W9. Paper convoluted.** It is not always easy to find all the important information when reading the paper the first time. For example, section F.2 is central to the understanding of the method and should therefore be in the main text.

---

> ### Author Rebuttal · Authors · 2026-03-31
>
> We thank the reviewer for their positive and professional evaluation and address the concerns below.
>
> >W1
>
> A1: We thank the reviewer for the suggestion. To ensure chemical distinctness, we filtered the solvation free energy test set using a Tanimoto similarity threshold of 0.65 [1] based on nearest-neighbor comparisons to the training set, computed with Morgan (ECFP4) fingerprints in RDKit. After filtering, 70 molecules remained, and the model was re-evaluated on this stricter split. The updated results are shown in [Fig A](https://anonymous.4open.science/r/CARD-88DE/materials/sfe.png). For the HiPen test set, the same analysis shows a maximum Tanimoto similarity of 0.43 to the training set, indicating that this split is already sufficiently distinct.
>
> >W2
>
> A2:  We thank the reviewer for the suggestion. We recomputed the total per-molecule cost including the generation of endpoint MD trajectories ([Table A](https://anonymous.4open.science/r/CARD-88DE/materials/duration.png)). Even with this cost included, CARD maintains an overall speedup of approximately 5× over MFES. Importantly, we note that endpoint MD is a shared prerequisite for nearly all free-energy methods. Moreover, our ablation study in **A7** shows that CARD is robust to the number and quality of reference structures and does not require fully converged MD trajectories, further preserving its practical inference-time advantage over traditional approaches.
>
> >W3
>
> A3:
> We thank the reviewer for the comment. Empirically, GAFF has an MAE of 1.11 kcal/mol and errors exceeding 10 kcal/mol on FreeSolv relative to experiment. Therefore, direct comparison to experiment would conflate force-field and model errors. To isolate method accuracy, we instead compare CARD and MFES under the same setup.
>
> >W4
>
> A4: We thank the reviewer for the valuable suggestion and have added overlap diagnostics on the vacuum-to-toluene solvation free energy task. For each test system, we compute the bidirectional effective sample sizes (ESS) between the CARD proposal (state 0) and the MD empirical distribution (state 1) in both vacuum and toluene, and take the minimum value across the two solvents for each direction (0→1 and 1→0). The relationship between the prediction error and ESS are shown in [Fig B](https://anonymous.4open.science/r/CARD-88DE/materials/overlap.png). We observe a Spearman correlation of -0.421 w.r.t. $\max(\text{ESS}_{01},\text{ESS}_{10})$, suggesting that this metric can serve as a practical indicator of potential failure modes.
>
> >W5
>
> A5: We thank the reviewer for the question. Due to space constraints, the discussion on SE(3) and PCA is provided in our response to Reviewer kmwG (Reply **A5**).
>
> >W6
>
> A6: We thank the reviewer for the comment. We have added experiments on the many-peptides-md dataset and compared CARD with Prose; details are provided in our response to Reviewer s8GM (Reply **A1**).
>
> >W7
>
> A7: We thank the reviewer for the helpful suggestion regarding additional ablations. We have now conducted the requested ablation studies, and the findings are summarized below.
> - [Table B](https://anonymous.4open.science/r/CARD-88DE/materials/ablation-ref.png). Model performance remains stable as long as more than one reference structure is used, indicating robustness to both the number and quality of reference structures.
> - [Table C](https://anonymous.4open.science/r/CARD-88DE/materials/ablation-arc.png). Topology-guided and distance-based atom orderings significantly outperform random ordering, with topology-guided ordering performing slightly better. Geometry-aware attention also provides a consistent performance improvement.
> - [Table D](https://anonymous.4open.science/r/CARD-88DE/materials/ablation-size.png). When trained on a random subset of 10k systems (out of 40,303), CARD's performance drops slightly, but the MAE remains below 1 kcal/mol, demonstrating robustness to reduced training data.
>
> >W8
>
> A8: We thank the reviewer for the observation. Performance can show sharp changes because $(r,L)$ take discrete values, so neighboring settings may induce substantially different distributions. As a potential remedy, one may adjust the continuous hyperparameter (e.g., the box size $a$) for smooth transitions between configurations. We will clarify this in the revised manuscript.
>
> >W9
>
> A9: We thank the reviewer for the suggestion. In the revised manuscript, we will move key methodological components (including Sec. F.2) to the main text for clarity.
>
> >Q1
>
> A10: Thanks for the question. Direct energy subtraction is generally ill-defined for MM force fields, as absolute energies are not comparable across difference topologies (no consistent formation-energy reference). Testing this baseline for the tautomer test set yields MAE = 21.68 kcal/mol, confirming that the thermodynamic cycle (using ANI-2x) is needed for accurate tautomer free-energy differences.
>
> **References**
>
> [1] Sayle, R. A. (2019). 2d similarity, diversity and clustering in rdkit. RDKit UGM.

---

> > ### Author Rebuttal · Reviewer_YsD4 · 2026-04-02
> >
> > The authors have addressed almost all of my concerns well.
> >
> > Only in regards to the question Q1, there was a misunderstanding:
> > I did not suggest to to estimate the change in tautomer free energy by taking differences in *potential* energies, which is of course ill-defined.
> > Rather, the authors currently estimate the change in tautomer free energy in water via the thermodynamic ∆Fw
> > 0→1 = ∆Fv0→1 +∆Fv→w(1) −∆Fv→w(0). This is certainly a sound approach. However, since CARD allows to estimate absolute *free* energies (based on the comparison to the 0-free energy reference), for both tautomers in water (Fw1 and Fw0), this begs the question why the change in free energy ∆Fw0→1  needs to be done in vacuum (∆Fv0→1) and corrected for by changes in hydration free energy rather than directly in (implicit solvent) water ∆Fw0→1 = Fw1 - Fw0.
> >
> > Edit: All my concerns have been addressed. I've updated my score accordingly.

---

> > > ### Author Response · Authors · 2026-04-03
> > >
> > > Dear Reviewer YsD4,
> > >
> > > Thank you for the clarification. We apologize again for the earlier misunderstanding.
> > >
> > > The main reason we did not directly compute $\Delta F_w^{0→1} = F_w^1 − F_w^0$ in implicit solvent is the need for a consistent free energy reference between two tautomers with different topologies. To directly compare absolute free energies across different topologies, the underlying potential must provide a consistent energy zero across molecules. Classical empirical force fields are not suitable for this purpose because their energy zero points are not consistent across molecules with different bonding topologies.
> > >
> > > In this work, we used the ANI-2x neural network potential, which is trained on DFT energies in the **gas phase** and therefore provides a consistent absolute energy reference in vacuum. However, we are not aware of a widely available transferable neural network potential trained on DFT with implicit solvent that could provide a consistent absolute free energy reference directly in solution. Because of this limitation, we adopted a thermodynamic cycle: we first compute the free energy difference in vacuum using a consistent reference, and then add solvation free energy corrections to obtain the free energy difference in water.
> > >
> > > We would like to emphasize that CARD itself can, in principle, be used to compute absolute free energies directly in implicit solvent. Our use of the thermodynamic cycle is therefore a practical choice due to the limitation of currently available potentials, rather than a limitation of the CARD framework.
> > >
> > > We hope this clarification addresses your concern. If you feel that the issue has been adequately resolved, we would appreciate your consideration in updating your evaluation accordingly. We also sincerely thank you for your constructive suggestions, which have helped make the paper more solid and complete.

---

### Official Review · Reviewer_wtgz · 2026-03-13

**Soundness:** 3
**Presentation:** 3
**Significance:** 3
**Originality:** 3
**Overall Recommendation:** 4
**Confidence:** 4

**Summary:**

The approach introduces CARD, a framework for estimating molecular free energy differences using an autoregressive model. The method converts 3D coordinates of small molecule conformations into mixed discrete-continuous sequences using a radix-based decomposition, enabling expressive coarse-to-fine autoregressive modelling. The approach is practically useful given its ability to estimate free energies without alchemical pathways unlike traditional MD. The experiments on ZINC20 show that it is capable of achieving accuracy comparable to classical computational methods on unseen molecular systems while providing significantly faster inference.

**Compliance With Llm Reviewing Policy:**

Affirmed.

**Final Justification:**

My primary concern was the quality of the benchmarking, which has been resolved with additional experiments on new datasets. Further, 2D torsion-torsion histograms were included to better demonstrate global performance, and the energy-minimization loss term was clearly explained and justified hence an increase in score.

**Key Questions For Authors:**

- Can the authors provide more clarity around how CARD outperforms DFT on the aqueous tautomer task?
- In a 10 ns MD simulation, are the small molecule conformations fully equilibrated? Is there a lot of flexibility in these molecules? Some sample Ramachandran plots to better demonstrate the conformational landscape should be included.
- Would it be possible to test the approach on a small peptide dataset, like ManyPeptidesMD (https://huggingface.co/datasets/transferable-samplers/many-peptides-md) to demonstrate scalability to more flexible systems?
- During training, was teacher-forcing used for evaluating the cross-entropy loss?
- The authors mention using sequence and context as conditioning, but the context used remains unclear?

**Limitations:**

Yes

**Strengths And Weaknesses:**

**Strengths**

- The approach to handling mixed discrete-continuous inputs using the radix decomposition is a great idea. The approach provides a bijective mapping between the discrete and continuous domains, which enables having a form of global conditioning for autoregressive generation.

- The experimental results are strong, using multiple solvents for analysis, and appear generalizable.


**Weaknesses**

- The conventional cross-entropy loss is great for optimization; however, the inclusion of the energy-based correction can significantly slow down training. The importance of the energy term on the quality of the results remains unclear. No ablations are conducted on $\lambda_1$ and $\lambda_2$ to investigate the relative importance of the cross-entropy term and the energy minimization term.

- It remains unclear how a model trained to minimize energies obtained from classic force fields outperforms DFT simulations, which should ideally serve as the ground truth labels for small molecule conformations. The current justification in the paper is not convincing.

- The benchmarking is lacking -- DFT and one other baseline are provided; however, many other works have been used for small molecule conformation sampling, and should be included to demonstrate the comparative performance gain by using CARD.

---

> ### Author Rebuttal · Authors · 2026-03-31
>
> We sincerely thank the reviewer for the careful reading and constructive comments, and we respond to the questions and identified weaknesses point by point below.
>
> > W1: About ablations on the energy minimization term.
>
> **A1**:
> We thank the reviewer for the suggestion regarding the energy-alignment term. We first clarify a potential misunderstanding: this term does not minimize predicted energies but ensures that their relative ordering and scale match the labels (i.e., lower-energy conformations under the force field correspond to lower predicted energies), without interfering with the cross-entropy objective.
>
> Its impact is evaluated in our ablation studies (Sections F.1 and G.1) of the original manuscript, using a two-stage training scheme: Stage I optimizes only cross-entropy, and Stage II incorporates both terms. As shown in Table 4, including the energy-alignment term consistently improves free-energy estimation. Given its importance, we will move these ablation results to the main text in the revised version.
>
> > W2: It remains unclear how a model trained to minimize energies obtained from classic force fields outperforms DFT simulations.
>
> **A2**:
> We appreciate the reviewer for raising this important question. While DFT is indeed the gold standard for *static electronic energy calculations*, the task in Section 4.3 is to compute the tautomer free energy , which depends on the full Boltzmann distribution of conformations rather than a single static structure.
>
> Since MD using DFT is computationally prohibitive, prior work [1] approximates the free energy difference by simply comparing the static DFT energies of the lowest-energy conformations (relaxed by DFT) of the two tautomers. This approximation assumes the probability mass is entirely concentrated at the energy minimum, which probably fails for flexible molecules.
>
> In contrast, CARD directly models the full Boltzmann distribution without relying on this minimum-energy assumption, avoiding the intrinsic errors of that approximation. This explains why CARD, despite using classical force-field energies, can outperform static DFT approximations for this specific task.
>
> To verify this, we grouped the test molecules by their number of rotatable bonds (Fig. 5). As expected, for rigid molecules (≤ 1 rotatable bond), CARD underperforms DFT. However, as molecular flexibility increases, CARD yields smaller errors and fewer outliers, strongly supporting our hypothesis.
>
> > Q1 & Q3: benchmark & test on ManyPeptidesMD.
>
> **A3**:
> We thank the reviewer for the suggestion. We supplement our study with experiments on the many-peptides-md dataset [4], comparing CARD with Prose [4], a Boltzmann generator that can generalize across peptide sequence lengths. Due to space limitations, we kindly refer the reviewer to our response to Reviewer s8GM (**A1**) for details.
>
> > Q2: Some sample Ramachandran plots.
>
> **A4**: We thank the reviewer for this question. Nanosecond-scale MD is commonly used for small-molecule simulations; for instance, [5] employs 5 ns production runs in explicit water to estimate hydration free energies.
>
> Following your suggestion, we randomly selected four training molecules and visualized their conformational landscapes using heavy-atom dihedral angles. For each molecule, dihedrals were ranked by circular variance, and 2D torsion-torsion histograms were plotted for the two most and two least mobile torsions (see [Fig A](https://anonymous.4open.science/r/CARD-88DE/materials/rama.png)). Even the least mobile torsions show some multimodality, indicating that 10 ns trajectories capture substantial conformational flexibility.
>
> > Q4: During training, was teacher-forcing used for evaluating the cross-entropy loss?
>
> **A5**: We thank the reviewer for the question. Teacher forcing is used when computing the cross-entropy loss during training. However, the coarse-to-fine autoregressive architecture mitigates the gap between teacher-forced training and autoregressive sampling, and we observe stable generation quality without noticeable out-of-distribution behavior.
>
> > Q5: The definition of *context*.
>
> **A6**: We thank the reviewer for the question. The definition of *context* is given in Section 3.1 and includes atomic numbers, covalent bond topology, and $R$ reference conformations sampled from the MD trajectory. We will revise the paper to make this definition clearer and more explicit.
>
> **References**
>
> [1] Pan, X. et al. (2025). Fast and accurate prediction of tautomer ratios in aqueous solution via a siamese neural network.
>
> [2] Klein, L. et al. (2024). Transferable boltzmann generators.
>
> [3] Du, Y. et al. (2025). FEAT: Free energy Estimators with Adaptive Transport.
>
> [4] Tan, C. B. et al. (2025). Amortized Sampling with Transferable Normalizing Flows.
>
> [5] Mobley, D. L. et al. (2009). Small molecule hydration free energies in explicit solvent: an extensive test of fixed-charge atomistic simulations.

---

> > ### Author Rebuttal · Reviewer_wtgz · 2026-04-04
> >
> > I would like to thank the authors for rigorously addressing my concerns. As a result, I'm happy to increase my score.

---

> > > ### Author Response · Authors · 2026-04-04
> > >
> > > Dear Reviewer wtgz,
> > >
> > > We sincerely thank the reviewer for the constructive feedback and for recognizing our efforts in addressing the concerns. We greatly appreciate your positive evaluation and your decision to increase the score.

---

### Decision · Program_Chairs · 2026-04-30

**Decision:**

Accept (regular)

**Comment:**

This paper introduces a novel approach using autoregressive models to estimate free energy differences. All reviewers agreed that the results are strong, and the fact that this approach can transfer across systems is very interesting.
The paper also introduces a host of clever ideas that are very compelling, in particular, the Radix decomposition is quite nice. Furthermore, the approach seems to provide strong empirical performance across multiple solvents.
Since this is the first AR model to be applied to this problem, I believe it is extremely important for the community to see this in place of existing flow-based approaches. Given that previous flow based methods applied to this problem suffer from poor scaling due to the intractibility of the likelihood, having a AR model which also gives exact likelihood but benefits from the vast sea of scaling literature in LLMs is a refreshing perspective to this long standing problem. Due to these significant merits, and new insights, I recommend acceptance of the paper.